# Generalizable spelling using a speech neuroprosthesis in an individual with severe limb and vocal paralysis

Sean L. Metzger [1,2,3,6], Jessie R. Liu [1,2,3,6], David A. Moses [1,2,6], Maximilian E. Dougherty[1], Margaret P. Seaton [1], Kaylo T. Littlejohn[1,2,4], Josh Chartier [1,2], Gopala K. Anumanchipalli[1,2,4], Adelyn Tu-Chan[5], Karunesh Ganguly [2,5] & Edward F. Chang [1,2,3] ✉

Neuroprostheses have the potential to restore communication to people who cannot speak or type due to paralysis. However, it is unclear if silent attempts to speak can be used to control a communication neuroprosthesis. Here, we translated direct cortical signals in a clinical-trial participant (Clinical-Trials.gov; NCT03698149) with severe limb and vocal-tract paralysis into single letters to spell out full sentences in real time. We used deep-learning and language-modeling techniques to decode letter sequences as the participant attempted to silently spell using code words that represented the 26 English letters (e.g. "alpha" for "a"). We leveraged broad electrode coverage beyond speech-motor cortex to include supplemental control signals from hand cortex and complementary information from low- and high-frequency signal components to improve decoding accuracy. We decoded sentences using words from a 1,152-word vocabulary at a median character error rate of 6.13% and speed of 29.4 characters per minute. In offline simulations, we showed that our approach generalized to large vocabularies containing over 9,000 words (median character error rate of 8.23%). These results illustrate the clinical viability of a silently controlled speech neuroprosthesis to generate sentences from a large vocabulary through a spelling-based approach, complementing previous demonstrations of direct full-word decoding.

Devastating neurological conditions such as stroke and amyotrophic lateral sclerosis can lead to anarthria, the loss of ability to communicate through speech[1]. Anarthric patients can have intact language skills and cognition, but paralysis may inhibit their ability to operate assistive devices, severely restricting communication with family, friends, and caregivers and reducing self-reported quality of life[2].

Brain-computer interfaces (BCIs) have the potential to restore communication to such patients by decoding neural activity into intended messages[3,4]. Existing communication BCIs typically rely on decoding imagined arm and hand movements into letters to enable spelling of intended sentences[5,6]. Although implementations of this approach have exhibited promising results, decoding natural attempts to speak directly into speech or text may offer faster and more natural

[1]Department of Neurological Surgery, University of California, San Francisco, San Francisco, CA, USA. [2]Weill Institute for Neuroscience, University of California, San Francisco, San Francisco, CA, USA. [3]University of California, Berkeley - University of California, San Francisco Graduate Program in Bioengineering, Berkeley, CA, USA. [4]Department of Electrical Engineering and Computer Sciences, University of California, Berkeley, Berkeley, CA, USA. [5]Department of Neurology, University of California, San Francisco, San Francisco, CA, USA. [6]These authors contributed equally: Sean L. Metzger, Jessie R. Liu, David A. Moses. ✉e-mail: edward.chang@ucsf.edu

control over a communication BCI. Indeed, a recent survey of prospective BCI users suggests that many patients would prefer speech-driven neuroprostheses over arm- and hand-driven neuroprostheses[7]. Additionally, there have been several recent advances in the understanding of how the brain represents vocal-tract movements to produce speech[8–11] and demonstrations of text decoding from the brain activity of able speakers[12–19], suggesting that decoding attempted speech from brain activity could be a viable approach for communication restoration.

To assess this, we recently developed a speech neuroprosthesis to directly decode full words in real time from the cortical activity of a person with anarthria and paralysis as he attempted to speak[20]. This approach exhibited promising decoding accuracy and speed, but as an initial study focused on a preliminary 50-word vocabulary. While direct word decoding with a limited vocabulary has immediate practical benefit, expanding access to a larger vocabulary of at least 1000 words would cover over 85% of the content in natural English sentences[21] and enable effective day-to-day use of assistive-communication technology[22]. Hence, a powerful complementary technology could expand current speech-decoding approaches to enable users to spell out intended messages from a large and generalizable vocabulary while still allowing fast, direct word decoding to express frequent and commonly used words. Separately, in this prior work the participant was controlling the neuroprosthesis by attempting to speak aloud, making it unclear if the approach would be viable for potential users who cannot produce any vocal output whatsoever.

Here, we demonstrate that real-time decoding of silent attempts to say 26 alphabetic code words from the NATO phonetic alphabet can enable highly accurate and rapid spelling in a clinical-trial participant (ClinicalTrials.gov; NCT03698149) with paralysis and anarthria. During training sessions, we cued the participant to attempt to produce individual code words and a hand-motor movement, and we used the simultaneously recorded cortical activity from an implanted 128-channel electrocorticography (ECoG) array to train classification and detection models. After training, the participant performed spelling tasks in which he spelled out sentences in real time with a 1152-word vocabulary using attempts to silently say the corresponding alphabetic code words. A beam-search algorithm used predicted code-word probabilities from a classification model to find the most likely sentence given the neural activity while automatically inserting spaces between decoded words. To initiate spelling, the participant silently attempted to speak, and a speech-detection model identified this start signal directly from ECoG activity. After spelling out the intended sentence, the participant attempted the hand-motor movement to disengage the speller. When the classification model identified this hand-motor command from ECoG activity, a large neural network-based language model rescored the potential sentence candidates from the beam search and finalized the sentence. In post-hoc simulations, our system generalized well across large vocabularies of over 9000 words.

## Results

### Overview of the real-time spelling pipeline

We designed a sentence-spelling pipeline that enabled a clinical-trial participant (ClinicalTrials.gov; NCT03698149) with anarthria and paralysis to silently spell out messages using signals acquired from a high-density electrocorticography (ECoG) array implanted over his sensorimotor cortex (Fig. 1). We tested the spelling system under copy-typing and conversational task conditions. In each trial of the copy-typing task condition, the participant was presented with a target sentence on a screen and then attempted to replicate that sentence (Supplementary Movie 1). In the conversational task condition, there were two types of trials: Trials in which the participant spelled out volitionally chosen responses to questions presented to him (Supplementary Movie 2) and trials in which he spelled out arbitrary,

unprompted sentences (Supplementary Movie 3). Prior to real-time testing, no day-of recalibration occured (Supplementary Movie 4); model parameters and hyperparameters were fit using data exclusively from preceding sessions.

When the participant was ready to begin spelling a sentence, he attempted to silently say an arbitrary word (Fig. 1a). We define silent-speech attempts as volitional attempts to articulate speech without vocalizing. Meanwhile, the participant's neural activity was recorded from each electrode and processed to simultaneously extract high-gamma activity (HGA; between 70 and 150 Hz) and low-frequency signals (LFS; between 0.3–100 Hz; Fig. 1b). A speech-detection model processed each time point of data in the combined feature stream (containing HGA+LFS features; Fig. 1c) to detect this initial silent-speech attempt.

Once an attempt to speak was detected, the paced spelling procedure began (Fig. 1d). In this procedure, an underline followed by three dots appeared on the screen in white text. The dots disappeared one by one, representing a countdown. After the last dot disappeared, the underline turned green to indicate a go cue, at which time the participant attempted to silently say the NATO code word corresponding to the first letter in the sentence. The time window of neural features from the combined feature stream obtained during the 2.5-s interval immediately following the go cue was passed to a neural classifier (Fig. 1e). Shortly after the go cue, the countdown for the next letter automatically started. This procedure was then repeated until the participant volitionally disengaged it (described later in this section).

The neural classifier processed each time window of neural features to predict probabilities across the 26 alphabetic code words (Fig. 1f). A beam-search algorithm used the sequence of predicted letter probabilities to compute potential sentence candidates, automatically inserting spaces into the letter sequences where appropriate and using a language model to prioritize linguistically plausible sentences. During real-time sentence spelling, the beam search only considered sentences composed of words from a predefined 1152-word vocabulary, which contained common words that are relevant for assistive-communication applications. The most likely sentence at any point in the task was always visible to the participant (Fig. 1d). We instructed the participant to continue spelling even if there were mistakes in the displayed sentence, since the beam search could correct the mistakes after receiving more predictions. After attempting to silently spell out the entire sentence, the participant was instructed to attempt to squeeze his right hand to disengage the spelling procedure (Fig. 1h). The neural classifier predicted the probability of this attempted hand-motor movement from each 2.5-s window of neural features, and if this probability was greater than 80%, the spelling procedure was stopped and the decoded sentence was finalized (Fig. 1i). To finalize the sentence, sentences with incomplete words were first removed from the list of potential candidates, and then the remaining sentences were rescored with a separate language model. The most likely sentence was then updated on the participant's screen (Fig. 1g). After a brief delay, the screen was cleared and the task continued to the next trial.

To train the detection and classification models prior to real-time testing, we collected data as the participant performed an isolated-target task. In each trial of this task, a NATO code word appeared on the screen, and the participant was instructed to attempt to silently say the code word at the corresponding go cue. In some trials, an indicator representing the hand-motor command was presented instead of a code word, and the participant was instructed to imagine squeezing his right hand at the go cue for those trials.

### Decoding performance

To evaluate the performance of the spelling system, we decoded sentences from the participant's neural activity in real time as he

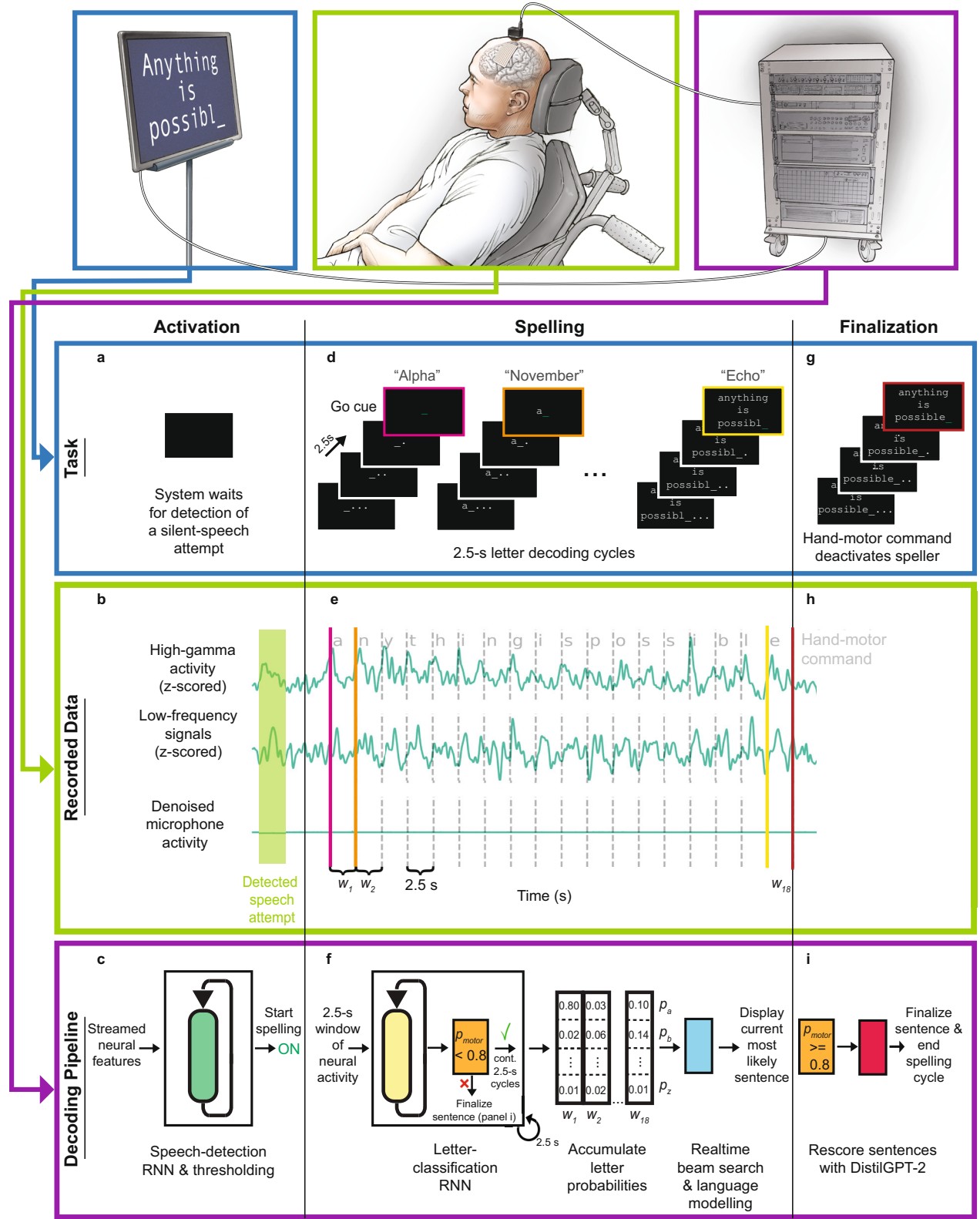

attempted to spell out 150 sentences (two repetitions each of 75 unique sentences selected from an assistive-communication corpus; see Table S1) during the copy-typing task. We evaluated the decoded sentences using word error rate (WER), character error rate (CER), words per minute (WPM), and characters per minute (CPM) metrics (Fig. 2). For characters and words, the error rate is defined as the edit distance, which is the minimum number of character or word deletions, insertions, and substitutions required to convert the predicted sentence to the target sentence that was displayed to the participant, divided by the total number of characters or words in the target sentence, respectively. These metrics are commonly used to assess the decoding performance of automatic speech recognition systems[23] and brain-computer interface applications[6,20].

We observed a median CER of 6.13% and median WER of 10.53% (99% confidence interval (CI) [2.25, 11.6] and [5.76, 24.8]) across the

**Fig. 1 | Schematic depiction of the spelling pipeline. a** At the start of a sentence-spelling trial, the participant attempts to silently say a word to volitionally activate the speller. **b** Neural features (high-gamma activity and low-frequency signals) are extracted in real time from the recorded cortical data throughout the task. The features from a single electrode (electrode 0, Fig. 5a) are depicted. For visualization, the traces were smoothed with a Gaussian kernel with a standard deviation of 150 milliseconds. The microphone signal shows that there is no vocal output during the task. **c** The speech-detection model, consisting of a recurrent neural network (RNN) and thresholding operations, processes the neural features to detect a silent-speech attempt. Once an attempt is detected, the spelling procedure begins. **d** During the spelling procedure, the participant spells out the intended message throughout letter-decoding cycles that occur every 2.5 s. Each cycle, the participant is visually presented with a countdown and eventually a go cue. At the go cue, the participant attempts to silently say the code word representing the desired letter. **e** High-gamma activity and low-frequency signals are computed throughout the

spelling procedure for all electrode channels and parceled into 2.5-s non-overlapping time windows. **f** An RNN-based letter-classification model processes each of these neural time windows to predict the probability that the participant was attempting to silently say each of the 26 possible code words or attempting to perform a hand-motor command (**g**). Prediction of the hand-motor command with at least 80% probability ends the spelling procedure (**i**). Otherwise, the predicted letter probabilities are processed by a beam-search algorithm in real time and the most likely sentence is displayed to the participant. **g** After the participant spells out his intended message, he attempts to squeeze his right hand to end the spelling procedure and finalize the sentence. **h** The neural time window associated with the hand-motor command is passed to the classification model. **i** If the classifier confirms that the participant attempted the hand-motor command, a neural network-based language model (DistilGPT-2) rescores valid sentences. The most likely sentence after rescoring is used as the final prediction.

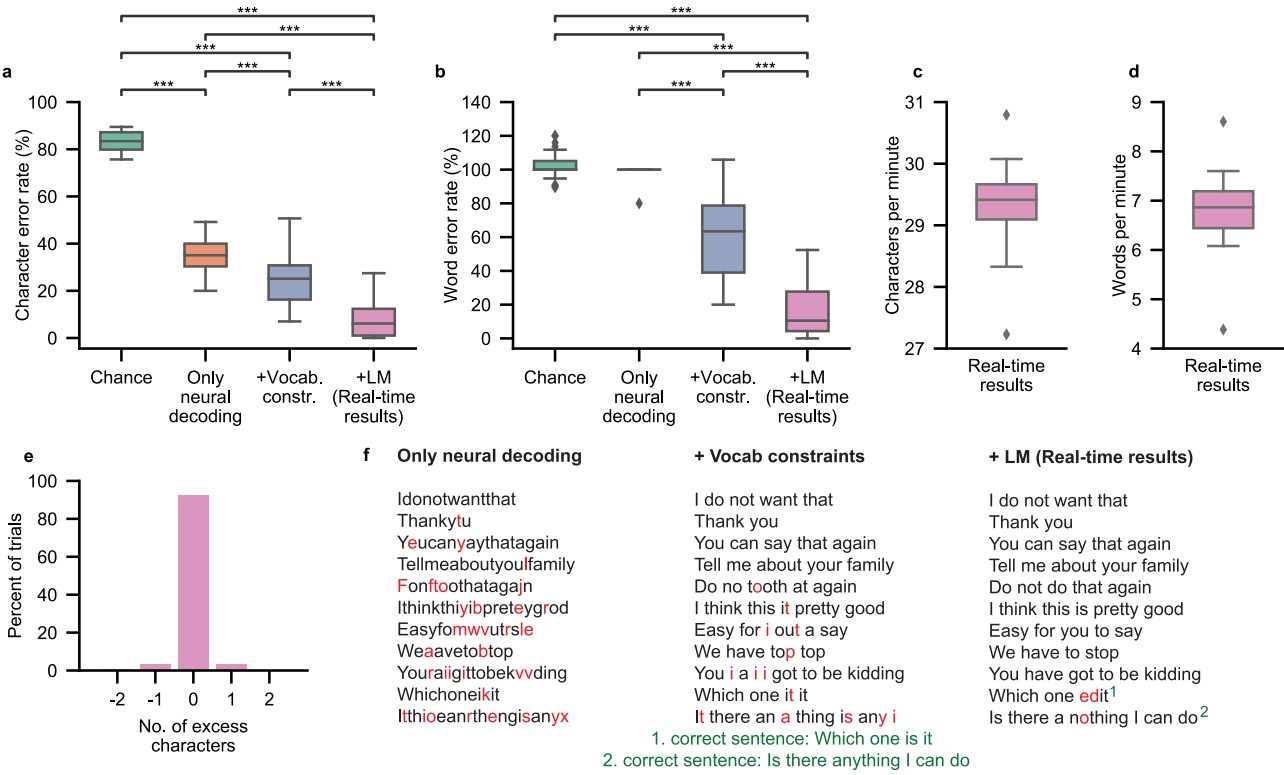

**Fig. 2 | Performance summary of the spelling system during the copy-typing task. a** Character error rates (CERs) observed during real-time sentence spelling with a language model (LM), denoted as '+LM (Real-time results)', and offline simulations in which portions of the system were omitted. In the 'Chance' condition, sentences were created by replacing the outputs from the neural classifier with randomly generated letter probabilities without altering the remainder of the pipeline. In the 'Only neural decoding' condition, sentences were created by concatenating together the most likely character from each of the classifier's predictions during a sentence trial (no whitespace characters were included). In the '+Vocab. constraints' condition, the predicted letter probabilities from the neural classifier were used with a beam search that constrained the predicted character sequences to form words within the 1152-word vocabulary. The final condition '+ LM (Real-time results)' incorporates language modeling. The sentences decoded with the full system in real time exhibited lower CERs than sentences decoded in the other conditions (***P < 0.0001, P-values provided in Table S2, two-sided Wilcoxon Rank-Sum test with 6-way Holm-Bonferroni correction). **b** Word error rates

(WERs) for real-time results and corresponding offline omission simulations from A (***P < 0.0001, P-values provided in Table S3, two-sided Wilcoxon Rank-Sum test with 6-way Holm-Bonferroni correction). **c** The decoded characters per minute during real-time testing. **d** The decoded words per minute during real-time testing. In **a**–**d**, the distribution depicted in each boxplot was computed across n = 34 real-time blocks (in each block, the participant attempted to spell between 2 and 5 sentences), and each boxplot depicts the median as a center line, quartiles as bottom and top box edges, and the minimum and maximum values as whiskers (except for data points that are 1.5 times the interquartile range, which are individually plotted). **e** Number of excess characters in each decoded sentence. **f** Example sentence-spelling trials with decoded sentences from each non-chance condition. Incorrect letters are colored red. Superscripts 1 and 2 denote the correct target sentence for the two decoded sentences with errors. All other example sentences did not contain any errors. Data to recreate panels **a**–**e** are provided as a Source Data file.

real-time test blocks (each block contained multiple sentence-spelling trials; Fig. 2a, b). Across 150 sentences, 105 (70%) were decoded without error, and 69 of the 75 sentences (92%) were decoded perfectly at least one of the two times they were attempted. Additionally, across 150 sentences, 139 (92.7%) sentences were decoded with the

correct number of letters, enabled by high classification accuracy of the attempted hand squeeze (Fig. 2e). We also observed a median CPM of 29.41 and median WPM of 6.86 (99% CI [29.1, 29.6] and [6.54, 7.12]) across test blocks, with spelling rates in individual blocks as high as 30.79 CPM and 8.60 WPM (Fig. 2c, d). These rates are higher than the

median rates of 17.37 CPM and 4.16 WPM (99% CI [16.1, 19.3] and [3.33, 5.05]) observed with the participant as he used his commercially available Tobii Dynavox assistive-typing device (as measured in our previous work[20]).

To understand the individual contributions of the classifier, beam search, and language model to decoding performance, we performed offline analyses using data collected during these real-time copy-typing task blocks (Fig. 2a, b). To examine the chance performance of the system, we replaced the model's predictions with randomly generated values while continuing to use the beam search and language model. This resulted in a CER and WER that was significantly worse than the real-time results ($z = 7.09$, $P = 8.08 \times 10^{-12}$ and $z = 7.09$, $P = 8.08 \times 10^{-12}$ respectively, two-sided Wilcoxon Rank-Sum test with 6-way Holm-Bonferroni correction). This demonstrates that the classification of neural signals was critical to system performance and that system performance was not just relying on a constrained vocabulary and language-modeling techniques.

To assess how well the neural classifier alone could decode the attempted sentences, we compared character sequences composed of the most likely letter for each individual 2.5-second window of neural activity (using only the neural classifier) to the corresponding target character sequences. All whitespace characters were ignored during this comparison (during real-time decoding, these characters were inserted automatically by the beam search). This resulted in a median CER of 35.1% (99% CI [30.6, 38.5]), which is significantly lower than chance ($z = 7.09$, $P = 8.08 \times 10^{-12}$, two-sided Wilcoxon Rank-Sum test with 6-way Holm-Bonferroni correction), and shows that time windows of neural activity during silent code-word production attempts were discriminable. The median WER was 100% (99% CI [100.0, 100.0]) for this condition; without language modeling or automatic insertion of whitespace characters, the predicted character sequences rarely matched the corresponding target character sequences exactly.

To measure how much decoding was improved by the beam search, we passed the neural classifier's predictions into the beam search and constrained character sequences to be composed of only words within the vocabulary without incorporating any language modeling. This significantly improved CER and WER over only using the most likely letter at each timestep ($z = 4.51$, $P = 6.37 \times 10^{-6}$ and $z = 6.61$, $P = 1.19 \times 10^{-10}$ respectively, two-sided Wilcoxon Rank-Sum test with 6-way Holm-Bonferroni correction). As a result of not using language modeling, which incorporates the likelihood of word sequences, the system would sometimes predict nonsensical sentences, such as "Do no tooth at again" instead of "Do not do that again" (Fig. 2f). Hence, including language modeling to complete the full real-time spelling pipeline significantly improved median CER to 6.13% and median WER to 10.53% over using the system without any language modeling ($z = 5.53$, $P = 6.34 \times 10^{-8}$ and $z = 6.11$, $P = 2.01 \times 10^{-9}$ respectively, two-sided Wilcoxon Rank-Sum test with 6-way Holm-Bonferroni correction), illustrating the benefits of incorporating the natural structure of English during decoding.

## Discriminatory content in high-gamma activity and low-frequency signals

Previous efforts to decode speech from brain activity have typically relied on content in the high-gamma frequency range (between 70 and 170 Hz, but exact boundaries vary) during decoding[12,13,24]. However, recent studies have demonstrated that low-frequency content (between 0 and 40 Hz) can also be used for spoken- and imagined-speech decoding[14,15,25–27], although the differences in the discriminatory information contained in each frequency range remain poorly understood.

In this work, we used both high-gamma activity (HGA; between 70 and 150 Hz) and low-frequency signals (LFS; between 0.3 and 16.67 Hz after downsampling with anti-aliasing) as neural features to enable sentence spelling. To characterize the speech content of each feature type, we used the most recent 10,682 trials of the isolated-target task) to train 10-fold cross-validated models using only HGA, only LFS, and both feature types simultaneously (HGA+LFS). In each of these trials, the participant attempted to silently say one of the 26 NATO code words. Models using only LFS demonstrated higher code-word classification accuracy than models using only HGA, and models using HGA+LFS out-performed the other two models ($z = 3.78$, $P = 4.71 \times 10^{-4}$ for all comparisons, two-sided Wilcoxon Rank-Sum test with 3-way Holm-Bonferroni correction; Figs. 3a and S4 and Table S4), achieving a median classification accuracy of 54.2% (99% CI [51.6, 56.2], Figs. 3a and S5). Confusion matrices depicting the classification results with each model are included in the supplement (Figs. S5–7).

We then investigated the relative contributions of each electrode and feature type to the neural classification models trained using HGA, LFS, and HGA+LFS. For each model, we first computed each electrode's contribution to classification by measuring the effect that small changes to the electrode's values had on the model's predictions[28]. Electrode contributions for the HGA model were primarily localized to the ventral portion of the grid, corresponding to the ventral aspect of the ventral sensorimotor cortex (vSMC), pars opercularis, and pars triangularis (Fig. 3b). Contributions for the LFS model were much more diffuse, covering more dorsal and posterior parts of the grid corresponding to dorsal aspects of the vSMC in the pre- and postcentral gyri (Fig. 3d). Contributions for the HGA model and the LFS model were moderately correlated with a Spearman rank correlation of 0.501 ($n = 128$ electrode contributions per feature type, $P < 0.01$). The separate contributions from HGA and LFS in the HGA+LFS model were highly correlated with the contributions for the HGA-only and LFS-only models, respectively ($n = 128$ electrode contributions per feature type, $P < 0.01$ for both Spearman rank correlations of 0.922 and 0.963, respectively; Fig. 3c, e). These findings indicate that the information contained in the two feature types that was most useful during decoding was not redundant and was recorded from relatively distinct cortical areas.

To further characterize HGA and LFS features, we investigated whether LFS had increased feature or temporal dimensionality, which could have contributed to increased decoding accuracy. First, we performed principal component analysis (PCA) on the feature dimension for the HGA, LFS, and HGA+LFS feature sets. The resulting principal components (PCs) captured the spatial variability (across electrode channels) for the HGA and LFS feature sets and the spatial and spectral variabilities (across electrode channels and feature types, respectively) for the HGA + LFS feature set. To explain more than 80% of the variance, LFS required significantly more feature PCs than HGA ($z = 12.2$, $P = 7.57 \times 10^{-34}$, two-sided Wilcoxon Rank-Sum test with 3-way Holm-Bonferroni correction; Fig. 3f) and the combined HGA+LFS feature set required significantly more feature PCs than the individual HGA or LFS feature sets ($z = 12.2$, $P = 7.57 \times 10^{-34}$ and $z = 11.6$, $P = 2.66 \times 10^{-33}$, respectively, two-sided Wilcoxon Rank-Sum test with 3-way Holm-Bonferroni correction; Fig. 3f). This suggests that LFS did not simply replicate HGA at each electrode but instead added unique feature variance.

To assess the temporal content of the features, we first used a similar PCA approach to measure temporal dimensionality. We observed that the LFS features required significantly more temporal PCs than both the HGA and HGA+LFS feature sets to explain more than 80% of the variance ($z = 12.2$, $P = 7.57 \times 10^{-34}$ and $z = 12.2$, $P = 7.57 \times 10^{-34}$, respectively, Fig. 3g; two-sided Wilcoxon Rank-Sum test with 3-way Holm-Bonferroni correction). Because the inherent temporal dimensionality for each feature type remained the same within the HGA+LFS feature set, the required number of temporal PCs to explain this much

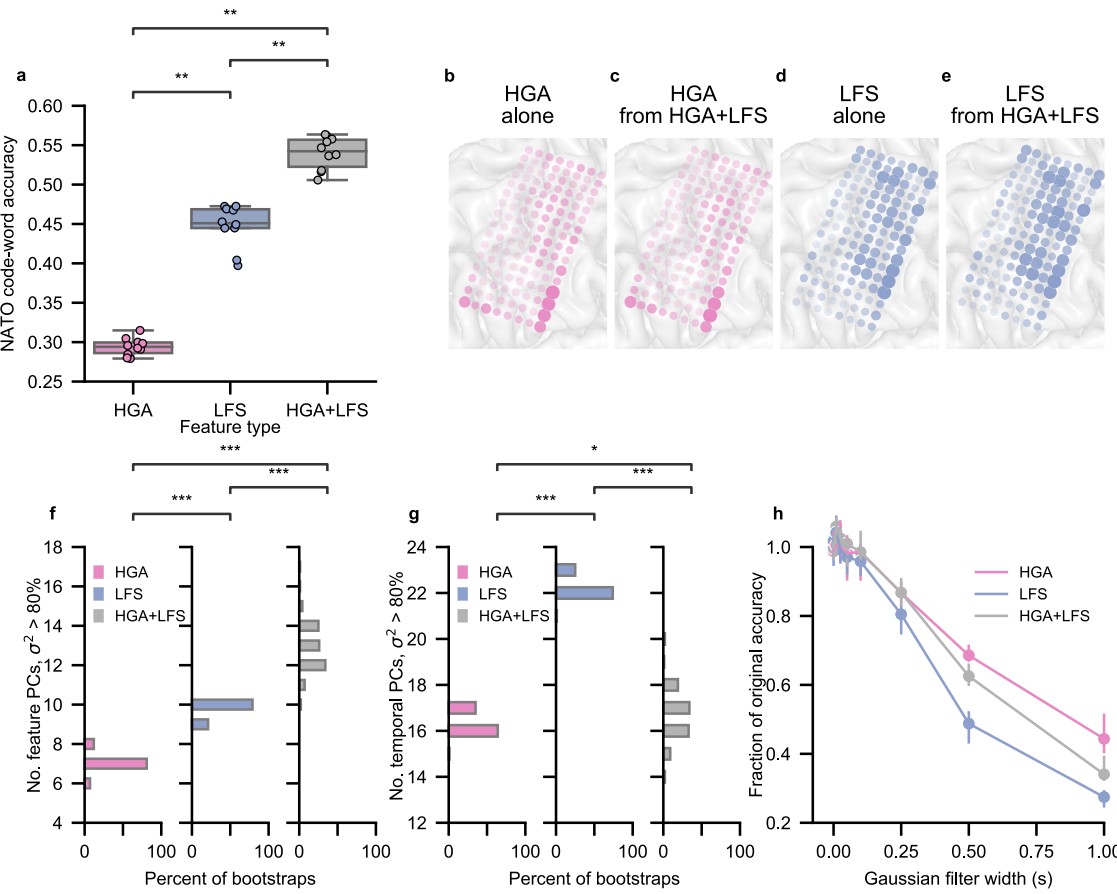

**Fig. 3 | Characterization of high-gamma activity (HGA) and low-frequency signals (LFS) during silent-speech attempts. a** 10-fold cross-validated classification accuracy on silently attempted NATO code words when using HGA alone, LFS alone, and both HGA+LFS simultaneously. Classification accuracy using only LFS is significantly higher than using only HGA, and using both HGA+LFS results in significantly higher accuracy than either feature type alone (**$P = 4.71 \times 10^{-4}$, $z = 3.78$ for each comparison, two-sided Wilcoxon Rank-Sum test with 3-way Holm-Bonferroni correction). Chance accuracy is 3.7%. Each boxplot corresponds to $n = 10$ cross-validation folds (which are also plotted as dots) and depicts the median as a center line, quartiles as bottom and top box edges, and the minimum and maximum values as whiskers (except for data points that are 1.5 times the interquartile range). **b–e** Electrode contributions. Electrodes that appear larger and more opaque provide more important features to the classification model. **b, c** Show contributions associated with HGA features using a model trained on HGA alone (**b**) vs using the combined LFS + HGA feature set (**c**). **d, e** depict contributions

associated with LFS features using a model trained on LFS alone (**d**) vs the combined LFS + HGA feature set (**e**). **f** Histogram of the minimum number of principal components (PCs) required to explain more than 80% of the total variance, denoted as $\sigma^2$, in the spatial dimension for each feature set over 100 bootstrap iterations. The number of PCs required were significantly different for each feature set (***$P < 0.0001$, $P$-values provided in Table S5, two-sided Wilcoxon Rank-Sum test with 3-way Holm-Bonferroni correction). **g** Histogram of the minimum number of PCs required to explain more than 80% of the variance in the temporal dimension for each feature set over 100 bootstrap iterations (***$P < 0.0001$, $P$-values provided in Table S6, two-sided Wilcoxon Rank-Sum test with 3-way Holm-Bonferroni correction, *$P < 0.01$ two-sided Wilcoxon Rank-Sum test with 3-way Holm-Bonferroni correction). **h** Effect of temporal smoothing on classification accuracy. Each point represents the median, and error bars represent the 99% confidence interval around bootstrapped estimations of the median. Data to recreate all panels are provided as a Source Data file.

variance for the HGA+LFS features was in between the corresponding numbers for the individual feature types. Then, to assess how the temporal resolution of each feature type affected decoding performance, we temporally smoothed each feature time series with Gaussian filters of varying widths. A wider Gaussian filter causes a greater amount of temporal smoothing, effectively temporally blurring the signal and hence lowering temporal resolution. Temporally smoothing the LFS features decreased the classification accuracy significantly more than smoothing the HGA or HGA+LFS features (Wilcoxon signed-rank statistic = 737.0, $P = 4.57 \times 10^{-5}$ and statistic = 391.0, $P = 1.13 \times 10^{-8}$, two-sided Wilcoxon signed-rank test with 3-way Holm-Bonferroni correction; Fig. 3h). The effects of temporal smoothing were not significantly different between HGA and HGA+LFS (Wilcoxon signed-rank statistic = 1460.0, $P = 0.443$). This is largely consistent with the outcomes of the temporal-PCA comparisons. Together, these results indicate that the temporal content of LFS had higher variability and contained more speech-related discriminatory information than HGA.

## Differences in neural discriminability between NATO code words and letters

During control of our system, the participant attempted to silently say NATO code words to represent each letter ("alpha" instead of "a", "beta" instead of "b", and so forth) rather than simply saying the letters themselves. We hypothesized that neural activity associated with attempts to produce code words would be more discriminable than letters due to increased phonetic variability and longer utterance lengths. To test this, we first collected data using a modified version of the isolated-target task in which the participant attempted to say each of the 26 English letters instead of the NATO code words that represented them. Afterwards, we trained and tested classification models using HGA+LFS features from the most recent 29 attempts to silently say each code word and each letter in 10-fold cross-validated analyses. Indeed, code words were classified with significantly higher accuracy than the letters ($z = 3.78$, $P = 1.57 \times 10^{-4}$, two-sided Wilcoxon Rank-Sum test; Fig. 4a).

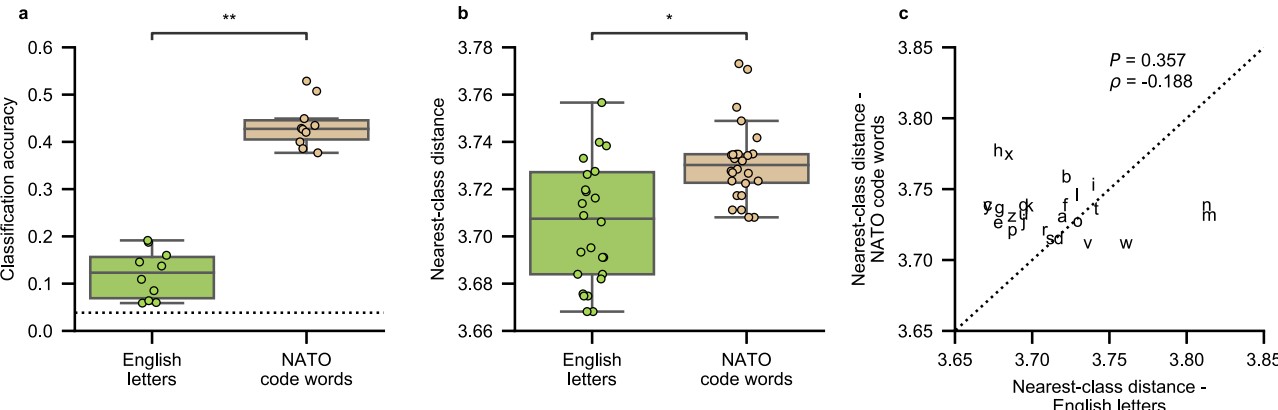

**Fig. 4 | Comparison of neural signals during attempts to silently say English letters and NATO code words. a** Classification accuracy (across $n = 10$ cross-validation folds) using models trained with HGA+LFS features is significantly higher for NATO code words than for English letters (**$P = 1.57 \times 10^{-4}$, $z = 3.78$, two-sided Wilcoxon Rank-Sum test). The dotted horizontal line represents chance accuracy. **b** Nearest-class distance is significantly larger for NATO code words than for letters (boxplots show values across the $n = 26$ code words or letters; *$P = 2.85 \times 10^{-3}$, $z = 2.98$, two-sided Wilcoxon Rank-Sum test). In **a**, **b**, each data point is plotted as a

dot, and each boxplot depicts the median as a center line, quartiles as bottom and top box edges, and the minimum and maximum values as whiskers (except for data points that are 1.5 times the interquartile range). **c** The nearest-class distance is greater for the majority of code words than for the corresponding letters. In **b** and **c**, nearest-class distances are computed as the Frobenius norm between trial-averaged HGA+LFS features. Data to recreate all panels are provided as a Source Data file.

To perform a model-agnostic comparison between the neural discriminability of each type of utterance (either code words or letters), we computed nearest-class distances for each utterance using the HGA+LFS feature set. Here, each utterance represented a single class, and distances were only computed between utterances of the same type. A larger nearest-class distance for a code word or letter indicates that that utterance is more discriminable in neural feature space because the neural activation patterns associated with silent attempts to produce it are more distinct from other code words or letters, respectively. We found that nearest-class distances for code words were significantly higher overall than for letters ($z = 2.98$, $P = 2.85 \times 10^{-3}$, two-sided Wilcoxon Rank-Sum test; Fig. 4b), although not all code words had a higher nearest-class distance than its corresponding letter (Fig. 4c).

### Distinctions in evoked neural activity between silent- and overt-speech attempts

The spelling system was controlled by silent-speech attempts, differing from our previous work in which the same participant used overt-speech attempts (attempts to speak aloud) to control a similar speech-decoding system[20]. To assess differences in neural activity and decoding performance between the two types of speech attempts, we collected a version of the isolated-target task in which the participant was instructed to attempt to say the code words aloud (overtly instead of silently). The spatial patterns of evoked neural activity for the two types of speech attempts exhibited similarities (Fig. S8), and inspections of evoked HGA for two electrodes suggest that some neural populations respond similarly for each speech type while others do not (Fig. 5a–c).

To compare the discriminatory neural content between silent- and overt-speech attempts, we performed 10-fold cross-validated classification analyses using HGA+LFS features associated with the speech attempts (Fig. 5d). First, for each type of attempted speech (silent or overt), we trained a classification model using data collected with that speech type. To determine if the classification models could leverage similarities in the neural representations associated with each speech type to improve performance, we also created models by pre-training on one speech type and then fine-tuning on the other speech type. We then tested each classification model on held-out data associated with each speech type and compared all 28 combinations of pairs of results (all statistical results detailed in Table S7). Models

trained solely on silent data but tested on overt data and vice versa resulted in classification accuracies that were above chance (median accuracies of 36.3%, 99% CI [35.0, 37.5] and 33.5%, 99% CI [31.0, 35.0], respectively; chance accuracy is 3.85%). However, for both speech types, training and testing on the same type resulted in significantly higher performance ($P < 0.01$, two-sided Wilcoxon Rank-Sum test, 28-way Holm-Bonferroni correction). Pre-training models using the other speech type led to increases in classification accuracy, though the increase was more modest and not significant for the overt speech type (median accuracy increasing by 2.33%, $z = 2.65$, $P = 0.033$ for overt, median accuracy increasing by 10.4%, $z = 3.78$, $P = 4.40 \times 10^{-3}$ for silent, two-sided Wilcoxon Rank-Sum test, 28-way Holm-Bonferroni correction). Together, these results suggest that the neural activation patterns evoked during silent and overt attempts to speak shared some similarities but were not identical.

### Generalizability to larger vocabularies and alternative tasks

Although the 1152-word vocabulary enabled communication of a wide variety of common sentences, we also assessed how well our approach can scale to larger vocabulary sizes. Specifically, we simulated the copy-typing spelling results using three larger vocabularies composed of 3303, 5249, and 9170 words that we selected based on their words' frequencies in large-scale English corpora. For each vocabulary, we retrained the language model used during the beam search to incorporate the new words. The large language model used when finalizing sentences was not altered for these analyses because it was designed to generalize to any English text.

High performance was maintained with each of the new vocabularies, with median character error rates (CERs) of 7.18% (99% CI [2.25, 11.6]), 7.93% (99% CI [1.75, 12.1]), and 8.23% (99% CI [2.25, 13.5]) for the 3303-, 5249-, and 9170-word vocabularies, respectively (Fig. 6a; median real-time CER was 6.13% (99% CI [2.25, 11.6]) with the original vocabulary containing 1,152 words). Median word error rates (WERs) were 12.4% (99% CI [8.01, 22.7]), 11.1% (99% CI [8.01, 23.1]), and 13.3% (99% CI [7.69, 28.3]), respectively (Fig. 6b; WER was 10.53% (99% CI [5.76, 24.8]) for the original vocabulary). Overall, no significant differences were found between the CERs or WERs with any two vocabularies ($P > 0.01$ for all comparisons, two-sided Wilcoxon Rank-Sum test with 6-way Holm-Bonferroni correction), illustrating the generalizability of our spelling approach to larger vocabulary sizes that enable fluent communication.

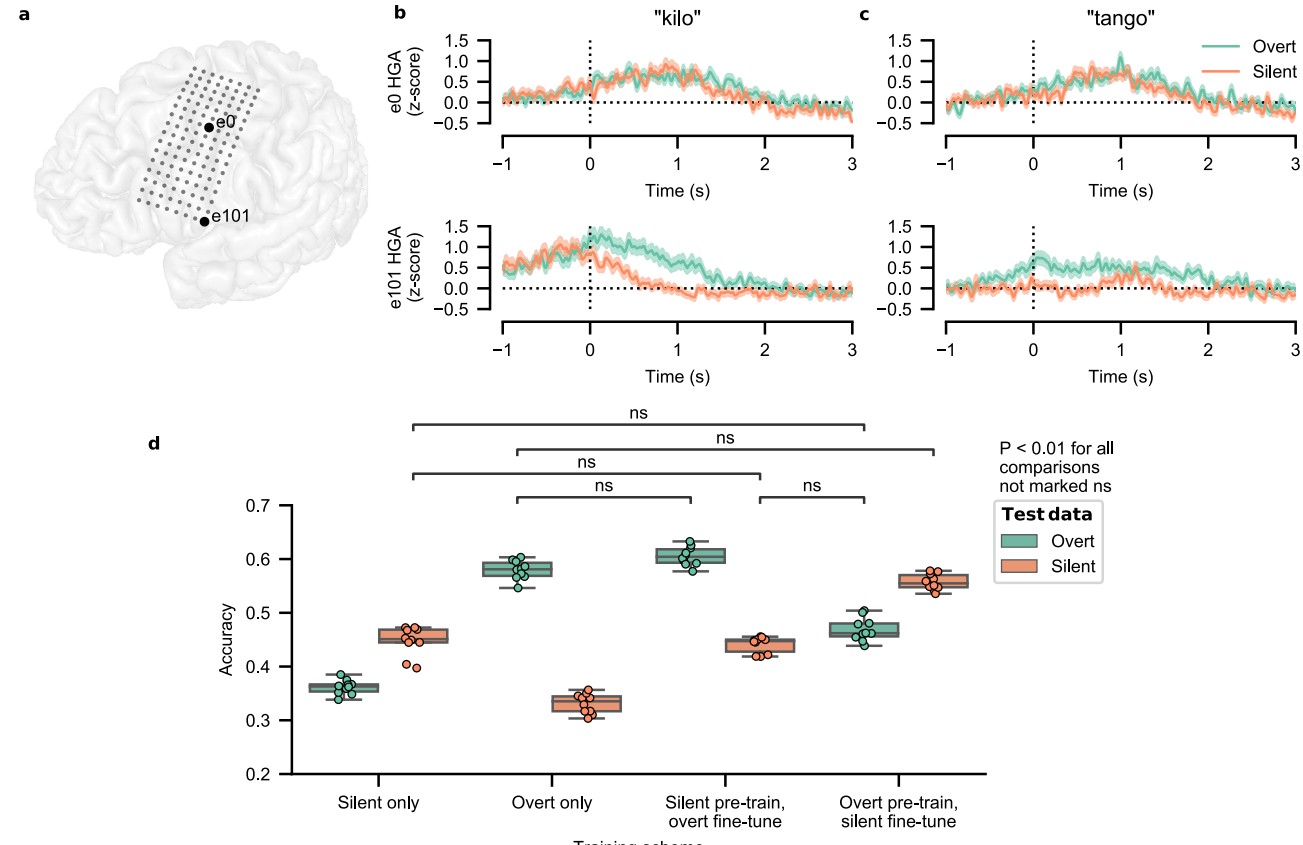

**Fig. 5 | Differences in neural signals and classification performance between overt- and silent-speech attempts. a** MRI reconstruction of the participant's brain overlaid with implanted electrode locations. The locations of the electrodes used in **b** and **c** are bolded and numbered in the overlay. **b** Evoked high-gamma activity (HGA) during silent (orange) and overt (green) attempts to say the NATO code word kilo. **c** Evoked high-gamma activity (HGA) during silent (orange) and overt (green) attempts to say the NATO code word tango. Evoked responses in **b** and **c** are aligned to the go cue, which is marked as a vertical dashed line at time 0. Each curve depicts the mean ± standard error across $n = 100$ speech attempts. **d** Code-word classification accuracy for silent- and overt-speech attempts with various model-training schemes. All comparisons revealed significant differences between the result pairs ($P < 0.01$, two-sided Wilcoxon Rank-Sum test with 28-way Holm-Bonferroni correction) except for those marked as 'ns'. Each boxplot corresponds to $n = 10$ cross-validation folds (which are also plotted as dots) and depicts the median as a center line, quartiles as bottom and top box edges, and the minimum and maximum values as whiskers (except for data points that are 1.5 times the interquartile range). Chance accuracy is 3.84%. Data to recreate all panels are provided as a Source Data file.

Finally, to assess the generalizability of our spelling approach to behavioral contexts beyond the copy-typing task structure, we measured performance as the participant engaged in a conversational task condition. In each trial of this condition, the participant was either presented with a question (as text on a screen) or was not presented with any stimuli. He then attempted to spell out a volitionally chosen response to the presented question (Supplementary Movie 2) or any arbitrary sentence if no stimulus was presented (Supplementary Movie 3). To measure the accuracy of each decoded sentence, we asked the participant to nod his head to indicate if the sentence matched his intended sentence exactly. If the sentence was not perfectly decoded, the participant used his commercially available assistive-communication device to spell out his intended message. Across 28 trials of this real-time conversational task condition, the median CER was 14.8% (99% CI [0.00, 29.7]) and the median WER was 16.7% (99% CI [0.00, 44.4]) (Fig. 6c, d). We observed a slight increase in decoding error rates compared to the copy-typing task, potentially due to the participant responding using incomplete sentences (such as "going out" and "summer time") that would not be well represented by the language models. Nevertheless, these results demonstrate that our spelling approach can enable a user to generate responses to questions as well as unprompted, volitionally chosen messages.

## Discussion

Here, we demonstrated that a paralyzed clinical-trial participant (ClinicalTrials.gov; NCT03698149) with anarthria could control a neuroprosthesis to spell out intended messages in real time using attempts to silently speak. With phonetically rich code words to represent individual letters and an attempted hand movement to indicate an end-of-sentence command, we used deep-learning and language-modeling techniques to decode sentences from electro-corticographic (ECoG) signals. These results significantly expand our previous word-decoding findings with the same participant[20] by enabling completely silent control, leveraging both high- and low-frequency ECoG features, including a non-speech motor command to finalize sentences, facilitating large-vocabulary sentence decoding through spelling, and demonstrating continued stability of the relevant cortical activity beyond 128 weeks since device implantation.

Previous implementations of spelling brain-computer interfaces (BCIs) have demonstrated that users can type out intended messages by visually attending to letters on a screen[29,30] or by using motor imagery to control a two-dimensional computer cursor[4,5] or attempt to handwrite letters[6]. BCI performance using penetrating microelectrode arrays in motor cortex has steadily improved over the past 20 years[31–33], recently achieving spelling rates as high as 90 characters per minute with a single participant[6], although this participant was able to speak normally. Our results extend the list of immediately practical

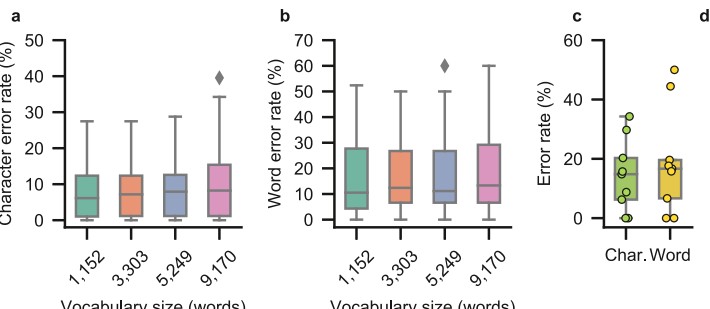

**Fig. 6 | The spelling approach can generalize to larger vocabularies and conversational settings. a** Simulated character error rates from the copy-typing task with different vocabularies, including the original vocabulary used during real-time decoding. **b** Word error rates from the corresponding simulations in **a**. In **a** and **b**, each boxplot corresponds to $n = 34$ blocks (in each of these blocks, the participant attempted to spell between two to five sentences). **c** Character and word error rates across the volitionally chosen responses and messages decoded in real time during the conversational task condition. Each boxplot corresponds to $n = 9$ blocks (in each of these blocks, the participant attempted to spell between two to four conversational responses; each dot corresponds to a single block). In **a**–**c**, each boxplot depicts the median as a center line, quartiles as bottom and top box edges, and the minimum and maximum values as whiskers (except for data points that are 1.5 times the interquartile range, which are individually plotted). **d** Examples of presented questions from trials of the conversational task condition (left) along with corresponding responses decoded from the participant's brain activity (right). In the final example, the participant spelled out his intended message without being prompted with a question. Data to recreate panels **a**–**c** are provided as a Source Data file.

and clinically viable control modalities for spelling-BCI applications to include silently attempted speech with an implanted ECoG array, which may be preferred for daily use by some patients due to the relative naturalness of speech[7] and may be more chronically robust across patients through the use of less invasive, non-penetrating electrode arrays with broader cortical coverage.

In post-hoc analyses, we showed that decoding performance improved as more linguistic information was incorporated into the spelling pipeline. This information helped facilitate real-time decoding with a 1152-word vocabulary, allowing for a wide variety of general and clinically relevant sentences as possible outputs. Furthermore, through offline simulations, we validated this spelling approach with vocabularies containing over 9000 common English words, which exceeds the estimated lexical-size threshold for basic fluency and enables general communication[34,35]. These results add to consistent findings that language modeling can significantly improve neural-based speech decoding[12,15,20] and demonstrates the immediate viability of speech-based spelling approaches for a general-purpose assistive-communication system.

In this study, we showed that neural signals recorded during silent-speech attempts by an anarthric person can be effectively used to drive a speech neuroprosthesis. Supporting the hypothesis that these signals contained similar speech-motor representations to signals recorded during overt-speech attempts, we showed that a model trained solely to classify overt-speech attempts can achieve above-chance classification of silent-speech attempts, and vice versa. Additionally, the spatial localization of electrodes contributing most to classification performance was similar for both overt and silent speech, with many of these electrodes located in the ventral sensorimotor cortex, a brain area that is heavily implicated in articulatory speech-motor processing[8–10,36].

Overall, these results further validate silently attempted speech as an effective alternative behavioral strategy to imagined speech and expand findings from our previous work involving the decoding of overt-speech attempts with the same participant[20], indicating that the production of residual vocalizations during speech attempts is not necessary to control a speech neuroprosthesis. These findings illustrate the viability of attempted-speech control for individuals with complete vocal-tract paralysis (such as those with locked-in syndrome), although future studies with these individuals are required to further our understanding of the neural differences between overt-speech attempts, silent-speech attempts, and purely imagined speech as well as how specific medical conditions might affect these

differences. We expect that the approaches described here, including recording methodology, task design, and modeling techniques, would be appropriate for both speech-related neuroscientific investigations and BCI development with patients regardless of the severity of their vocal-tract paralysis, assuming that their speech-motor cortices are still intact and that they are mentally capable of attempting to speak.

In addition to enabling spatial coverage over the lateral speech-motor cortical brain regions, the implanted ECoG array also provided simultaneous access to neural populations in the hand-motor (hand knob) cortical area that is typically implicated during executed or attempted hand movements[37]. Our approach is the first to combine the two cortical areas to control a BCI. This ultimately enabled our participant to use an attempted hand movement, which was reliably detectable and highly discriminable from silent-speech attempts with 98.43% classification accuracy (99% CI [95.31, 99.22]), to indicate when he was finished spelling any particular sentence. This may be a preferred stopping mechanism compared to previous spelling BCI implementations that terminated spelling for a sentence after a pre-specified time interval had elapsed or extraneously when the sentence was completed[5] or required a head movement to terminate the sentence[6]. By also allowing a silent-speech attempt to initiate spelling, the system could be volitionally engaged and disengaged by the participant, which is an important design feature for a practical communication BCI. Although attempted hand movement was only used for a single purpose in this first demonstration of a multimodal communication BCI, separate work with the same participant suggests that non-speech motor imagery could be used to indicate several distinct commands[38].

One drawback of the current approach is that it relies on code words instead of letters during spelling. Although the use of these longer code words improved neural discriminability, they are less natural to use. Separately, the participant had to attempt to produce code words at a pre-defined pace during spelling, which enabled straightforward parcellation of the neural activity into separate time windows for classification but reduced flexibility for the user. Future work can focus on improving letter decoding and implementing flexible, user-controlled pacing (for example, through augmented speech-attempt detection) to facilitate more naturalistic spelling. Additionally, the present results are limited to only one participant; to fully assess the clinical viability of this spelling system as a neuroprosthesis, it will need to be validated with more participants.

In future communication neuroprostheses, it may be possible to use a combined approach that enables rapid decoding of full words or

phrases from a limited, frequently used vocabulary[20] as well as slower, generalizable spelling for out-of-vocabulary items. Transfer-learning methods could be used to cross-train differently purposed speech models using data aggregated across multiple tasks and vocabularies, as validated in previous speech-decoding work[13]. Although clinical and regulatory guidelines concerning the implanted percutaneous connector prevented the participant from being able to use the current spelling system independently, development of a fully implantable ECoG array and a software application to integrate the decoding pipeline with an operating system's accessibility features could allow for autonomous usage. Facilitated by deep-learning techniques, language modeling, and the signal stability and spatial coverage afforded by ECoG recordings, future communication neuroprostheses could enable users with severe paralysis and anarthria to control assistive technology and personal devices using naturalistic silent-speech attempts to generate intended messages and attempted non-speech motor movements to issue high-level, interactive commands.

## Methods

### Clinical trial overview
This study was conducted as part of the BCI Restoration of Arm and Voice (BRAVO) clinical trial (ClinicalTrials.gov; NCT03698149). The goal of this single-institution clinical trial is to assess the incidence of treatment-emergent adverse events associated with the ECoG-based neural interface and to determine if ECoG and custom decoding methods can enable long-term assistive neurotechnology to restore communication and mobility. The data presented here and the present work do not support or inform any conclusions about the primary outcomes of this trial. The clinical trial began in November 2018. The Food and Drug Administration approved an investigational device exemption for the neural implant used in this study. The study protocol was approved by the Committee on Human Research at the University of California, San Francisco. The data safety monitoring board agreed to the release of the results of this work prior to the completion of the trial. The participant gave his informed consent to participate in this study after the details concerning the neural implant, experimental protocols, and medical risks were thoroughly explained to him. The full clinical-trial protocol, along with a note that frames the present work within the exploratory clinical trial, can be found as a supplementary file alongside the online version of this article.

### Participant
The participant, who was 36 years old at the start of the study, was diagnosed with severe spastic quadriparesis and anarthria by neurologists and a speech-language pathologist after experiencing an extensive pontine stroke (Note S1). He is fully cognitively intact. Although he retains the ability to vocalize grunts and moans, he is unable to produce intelligible speech, and his attempts to speak aloud are abnormally effortful due to his condition (according to self-reported descriptions; Note S2). He typically relies on assistive computer-based interfaces that he controls with residual head movements to communicate. This participant has participated in previous studies as part of this clinical trial[20,38], although neural data from those studies were not used in the present study. He provided verbal consent (using his assistive computer-based interface) to participate in the study and allow his image to appear in the supplementary videos accompanying this article. He also provided verbal consent (again using this interface) to have a designated third-party individual physically sign the consent forms on his behalf.

### Neural implant
The neural implant device consisted of a high-density electrocorticography (ECoG) array (PMT) and a percutaneous connector (Blackrock Microsystems)[20]. The ECoG array contained 128 disk-shaped electrodes arranged in a lattice formation with 4-mm center-

to-center spacing. The array was surgically implanted on the pial surface of the left hemisphere of the brain over cortical regions associated with speech production, including the dorsal posterior aspect of the inferior frontal gyrus, the posterior aspect of the middle frontal gyrus, the precentral gyrus, and the anterior aspect of the postcentral gyrus[8,10,39]. The percutaneous connector was implanted in the skull to conduct electrical signals from the ECoG array to a detachable digital headstage and cable (NeuroPlex E; Blackrock Microsystems), minimally processing and digitizing the acquired brain activity and transmitting the data to a computer. The device was implanted in February 2019 at UCSF Medical Center without any surgical complications.

### Data acquisition and preprocessing
We acquired neural features from the implanted ECoG array using a pipeline involving several hardware components and processing steps (Fig. S2). We connected a headstage (a detachable digital connector; NeuroPlex E, Blackrock Microsystems) to the percutaneous pedestal connector, which digitized neural signals from the ECoG array and transmitted them through an HDMI connection to a digital hub (Blackrock Microsystems). The digital hub then transmitted the digitized signals through an optical fiber cable to a Neuroport system (Blackrock Microsystems), which applied noise cancellation and an anti-aliasing filter to the signals before streaming them at 1 kHz through an Ethernet connection to a separate real-time computer (Colfax International). The Neuroport system was controlled using the NeuroPort Central Suite software package (version 7.0.4; Blackrock Microsystems).

On the real-time processing computer, we used a custom Python software package (rtNSR) to process and analyze the ECoG signals, execute the real-time tasks, perform real-time decoding, and store the data and task metadata[20,24,40]. Using this software package, we first applied a common average reference (across all electrode channels) to each time sample of the ECoG data. Common average referencing is commonly applied to multi-channel datasets to reduce shared noise[41,42]. These re-referenced signals were then processed in two parallel processing streams to extract high-gamma activity (HGA) and low-frequency signal (LFS) features using digital finite impulse response (FIR) filters designed using the Parks-McClellan algorithm[43] (Fig. S2; filters were designed using the SciPy Python package (version 1.5.4)). Briefly, we used these FIR filters to compute the analytic amplitude of the signals in the high-gamma frequency band (70–150 Hz) and an anti-aliased version of the signals (with a cutoff frequency at 100 Hz). We combined the time-synchronized high-gamma analytic amplitudes and downsampled signals into a single feature stream at 200 Hz. Next, we z-scored the values for each channel and each feature type using a 30-s sliding window to compute running statistics. Finally, we implemented an artifact-rejection approach that identified neural time points containing at least 32 features with z-score magnitudes greater than 10, replacing each of these time points with the z-score values from the preceding time point and ignoring these time points when updating the running z-score statistics. During real-time decoding and in offline analyses, we used the z-scored high-gamma analytic amplitudes as the HGA features and the z-scored downsampled signals as the LFS features (and the combination of the two as the HGA+LFS feature set). The neural classifier further downsampled these feature streams by a factor of 6 before using them for inference (using an anti-aliasing filter with a cutoff frequency at 16.67 Hz), but the speech detector did not.

We performed all data collection and real-time decoding tasks in the participant's bedroom or a small office room nearby. We uploaded data to our lab's server infrastructure and trained the decoding models using NVIDIA V100 GPUs hosted on this infrastructure.

## Task design

We recorded neural data with the participant during two general types of tasks: an isolated-target task and a sentence-spelling task (Fig. S1). In each trial of the isolated-target task, a text target appeared on the screen along with 4 dots on either side. Dots on both sides disappeared one by one until no dots remained, at which point the text target turned green to represent a go cue. At this go cue, the participant either attempted to say the target (silently or aloud, depending on the current task instructions) if it was either a NATO code word or an English letter. If the target was a text string containing the word "Right" and an arrow pointing right, the participant instead attempted to squeeze his right hand. We used the neural data collected during the isolated-target task to train and optimize the detection and classification models and to evaluate classifier performance (Method S1 and Note S3).

The sentence-spelling task is described in the start of the Results section and in Fig. 1. Briefly, the participant used the full spelling pipeline (described in the following sub-section) to either spell sentences presented to him as targets in a copy-typing task condition or to spell arbitrary sentences in a conversational task condition. We did not implement functionality to allow the participant to retroactively alter the predicted sentence, although the language model could alter previously predicted words in a sentence after receiving additional character predictions. Data collected during the sentence-spelling task were used to optimize beam-search hyperparameters and evaluate the full spelling pipeline.

## Modeling

We fit detection and classification models using data collected during the isolated-target task as the participant attempted to produce code words and the hand-motor command. After fitting these models offline, we saved the trained models to the real-time computer for use during real-time testing. We implemented these models using the PyTorch Python package (version 1.6.0). In addition to these two models, we also used language models to enable sentence spelling. We used hyperparameter optimization procedures on held-out validation datasets to choose values for model hyperparameters (see Table S8). We used the Python software packages NumPy (version 1.19.1), scikit-learn (version 0.24.2), and pandas (version 0.25.3) during modeling and data analysis.

**Speech detection.** To determine when the participant was attempting to engage the spelling system, we developed a real-time silent-speech detection model. This model used long short-term memory layers, a type of recurrent neural network layer, to process neural activity in real time and detect attempts to silently speak[20]. This model used both LFS and HGA features (a total of 256 individual features) at 200 Hz.

The speech-detection model was trained using supervised learning and truncated backpropagation through time. For training, we labeled each time point in the neural data as one of four classes depending on the current state of the task at that time: 'rest', 'speech preparation', 'motor', and 'speech.' Though only the speech probabilities were used during real-time evaluation to engage the spelling system, the other labels were included during training to help the detection model disambiguate attempts to speak from other behavior. See Method S2 and Fig. S3 for further details about the speech-detection model.

**Classification.** We trained an artificial neural network (ANN) to classify the attempted code word or hand-motor command $y_i$ from the time window of neural activity $x_i$ associated with an isolated-target trial or 2.5-s letter-decoding cycle $i$. The training procedure was a form of maximum likelihood estimation, where given an ANN classifier parameterized by $\theta$ and conditioned on the neural activity $x_i$, our goal during model fitting was to find the parameters $\theta^*$ that maximized the probability of the training labels. This can be written as the following optimization problem:

$$\theta^* = \arg\max_\theta \prod_i p_\theta(y_i|x_i) = \arg\min_\theta - \sum_i \log p_\theta(y_i|x_i) \qquad (1)$$

We approximated the optimal parameters $\theta^*$ using stochastic gradient descent and the Adam optimizer[44].

To model the temporal dynamics of the neural time-series data, we used an ANN with a one-dimensional temporal convolution on the input layer followed by two layers of bidirectional gated recurrent units (GRUs)[45], for a total of three layers. We multiplied the final output of the last GRU layer by an output matrix and then applied a softmax function to yield the estimated probability of each of the 27 labels $\hat{y}_i$ given $x_i$. See Method S3 for further details about the data-processing, data-augmentation, hyperparameter-optimization, and training procedures used to fit the neural classifier.

**Classifier ensembling for sentence spelling.** During sentence spelling, we used model ensembling to improve classification performance by reducing overfitting and unwanted modeling variance caused by random parameter initializations[46]. Specifically, we trained 10 separate classification models using the same training dataset and model architecture but with different random parameter initializations. Then, for each time window of neural activity $x_i$, we averaged the predictions from these 10 different models together to produce the final prediction $\hat{y}_i$.

**Incremental classifier recalibration for sentence spelling.** To improve sentence-spelling performance, we trained the classifiers used during sentence spelling on data recorded during sentence-spelling tasks from preceding sessions (in addition to data from the isolated-target task). In an effort to only include high-quality sentence-spelling data when training these classifiers, we only used data from sentences that were decoded with a character error rate of 0.

**Beam search.** During sentence spelling, our goal was to compute the most likely sentence text $s^*$ given the neural data $X$. We used the formulation from Hannun et al[23]. to find $s^*$ given its likelihood from the neural data and its likelihood under an adjusted language-model prior, which allowed us to incorporate word-sequence probabilities with predictions from the neural classifier. This can be expressed formulaically as:

$$s^* = \arg\max_s p_{nc}(s|X) p_{lm}(s^\alpha)|s|^\beta \qquad (2)$$

Here, $p_{nc}(s|X)$ is the probability of $s$ under the neural classifier given each window of neural activity, which is equal to the product of the probability of each letter in $s$ given by the neural classifier for each window of neural activity $x_i$. $p_{lm}(s)$ is the probability of the sentence $s$ under a language-model prior. Here, we used an n-gram language model to approximate $p_{lm}(s)$. Our n-gram language model, with $n = 3$, provides the probability of each word given the preceding two words in a sentence. We implemented this language model using custom code as well as utility functions from the NLTK Python package (version 3.6.2). The probability under the language model of a sentence is then taken as the product of the probability of each word given the two words that precede it (Method S4).

As in Hannun et al.[23], we assumed that the n-gram language-model prior was too strong and downweighted it using a hyperparameter $\alpha$. We also included a word-insertion bonus $\beta$ to encourage the language model to favor sentences containing more words, counteracting an implicit consequence of the language model that causes the probability of a sentence under it $p_{lm}(s)$ to decrease as the number of words in $s$ increases. $|s|$ denotes the cardinality of $s$, which is equal to the

number of words in *s*. If a sentence *s* was partially completed, only the words preceding the final whitespace character in *s* were considered when computing $p_{lm}(s)$ and $|s|$.

We then used an iterative beam-search algorithm as in Hannun et al.[23] to approximate $s^*$ at each timepoint $t = \tau$. We used a list of the $B$ most likely sentences from $t = \tau-1$ (or a list containing a single empty-string element if $t = 1$) as a set of candidate prefixes, where $B$ is the beam width. Then, for each candidate prefix $l$ and each English letter $c$ with $p_{nc}(c|x_\tau) > 0.001$, we constructed new candidate sentences by considering $l$ followed by $c$. Additionally, for each candidate prefix $l$ and each text string $c^+$, composed of an English letter followed by the whitespace character, with $p_{nc}(c^+|x_\tau) > 0.001$, we constructed more new candidate sentences by considering $l$ followed by $c^+$. Here and throughout the beam search, we considered $p_{nc}(c^+|x_\tau) = p_{nc}(c|x_\tau)$ for each $c$ and corresponding $c^+$. Next, we discarded any resulting candidate sentences that contained words or partially completed words that were not valid given our constrained vocabulary. Then, we rescored each remaining candidate sentence $\tilde{l}$ with $p(\tilde{l}) = p_{nc}(\tilde{l}|X_{1:\tau})p_{lm}(\tilde{l})^\alpha |\tilde{l}|^\beta$. The most likely candidate sentence, $s^*$, was then displayed as feedback to the participant

We chose values for $\alpha$, $\beta$, and $B$ using hyperparameter optimization (see Method S5 for more details).

If at any time point $t$ the probability of the attempted hand-motor command (the sentence-finalization command) was >80%, the $B$ most likely sentences from the previous iteration of the beam search were processed to remove any sentence with incomplete or out-of-vocabulary words. The probability of each remaining sentence $\hat{l}$ was then recomputed as:

$$p(\hat{l}) = p_{nc}(\hat{l}|X_{1:t-1})p_{lm}(\hat{l})^\alpha |\hat{l}|^\beta p_{gpt2}(\hat{l})^{\alpha_{gpt2}} \qquad (3)$$

Here, $p_{gpt2}(\hat{l})$ denotes the probability of $\hat{l}$ under the DistilGPT-2 language model, a low-parameter variant of GPT-2 (implemented using the lm-scorer Python package (version 0.4.2); see Method S4 for more details), and $\alpha_{gpt2}$ represents a scaling hyperparameter that was set through hyperparameter optimization. The most likely sentence $\hat{l}$ given this formulation was then displayed to the participant and stored as the finalized sentence.

See Method S5 for further details about the beam-search algorithm.

## Performance evaluation

**Character error rate and word error rate.** Because CER and WER are overly influenced by short sentences, as in previous studies[6,20] we reported CER and WER as the sum of the character or word edit distances between each of the predicted and target sentences in a sentence-spelling block and then divided this number by the total number of characters or words across all target sentences in the block. Each block contained between two to five sentence trials.

**Assessing performance during the conversational task condition.** To obtain ground-truth sentences to calculate CERs and WERs for the conversational condition of the sentence-spelling task, after completing each block we reminded the participant of the questions and the decoded sentences from that block, and then, for each decoded sentence, he either confirmed that the decoded sentence was correct or typed out the intended sentence using his commercially available assistive-communication device. Each block contained between two to four sentence trials.

**Characters and words per minute.** We calculated the characters per minute and words per minute rates for each sentence-spelling (copy-typing) block as follows:

$$\text{rate} = \frac{\sum_i N_i}{\sum_i D_i} \qquad (4)$$

Here, $i$ indexes each trial, $N_i$ denotes the number of words or characters (including whitespace characters) decoded for trial $i$, and $D_i$ denotes the duration of trial $i$ (in minutes; computed as the difference between the time at which the window of neural activity corresponding to the final code word in trial $i$ ended and the time of the go cue of the first code word in trial $i$).

**Electrode contributions.** To compute electrode contributions using data recorded during the isolated-target task, we computed the derivative of the classifier's loss function with respect to the input features across time as in Simonyan et al.[28], yielding a measure of how much the predicted model outputs were affected by small changes to the input feature values for each electrode and feature type (HGA or LFS) at each time point. Then, we calculated the L2-norm of these values across time and averaged the resulting values across all isolated-target trials, yielding a single contribution value for each electrode and feature type for that classifier.

**Cross-validation.** For each fold, we used stratified cross-validation folds of the isolated-target task. We split each fold into a training set containing 90% of the data and a held-out testing set containing the remaining 10%. In all, 10% of the training dataset was then randomly selected (with stratification) as a validation set.

**Analyzing neural-feature principal components.** To characterize the HGA and LFS neural features, we used bootstrapped principal component analyses. First, for each NATO code word, we randomly sampled (with replacement) cue-aligned time windows of neural activity (spanning from the go cue to 2.5 s after the go cue) from the first 318 silently attempted isolated-target trials for that code word. To clearly understand the role of each feature stream for classification, we downsampled the signals by a factor of 6 to obtain the signals used by the classifier. Then, we trial averaged the data for each code word, yielding 26 trial averages across time for each electrode and feature set (HGA, LFS, and HGA+LFS). We then arranged this into a matrix with dimensionality $N \times TC$, where $N$ is the number of features (128 for HGA and for LFS; 256 for HGA+LFS), $T$ is the number of time points in each 2.5-s window, and $C$ is the number of NATO code words (26), by concatenating the trial-averaged activity for each feature. We then performed principal component analysis along the feature dimension of this matrix. Additionally, we arranged the trial-averaged data for each code word into a matrix with dimensionality $T \times NC$. We then performed principal component analysis along the temporal dimension. For each analysis, we performed the measurement procedure 100 times to obtain a representative distribution of the minimum number of principal components required to explain more than 80% of the variance.

**Nearest-class distance comparison.** To compare nearest-class distances for the code words and letters, we first calculated averages across 1000 bootstrap iterations of the combined HGA+LFS feature set across 47 silently attempted isolated-target trials for each code word and letter. We then computed the Frobenius norm of the difference between each pairwise combination. For each code word, we used the smallest computed distance between that code word and any other code word as the nearest-class distance. We then repeated this process for the letters.

**Generalizability to larger vocabularies.** During real-time sentence spelling, the participant created sentences composed of words from a

1152-word vocabulary that contained common words and words relevant to clinical caregiving. To assess the generalizability of our system, we tested the sentence-spelling approach in offline simulations using three larger vocabularies. The first of these vocabularies was based on the 'Oxford 3000' word list, which is composed of 3000 core words chosen based on their frequency in the Oxford English Corpus and relevance to English speakers[47]. The second was based on the 'Oxford 5000' word list, which is the 'Oxford 3000' list augmented with an additional 2,000 frequent and relevant words. The third was a vocabulary based on the most frequent 10,000 words in Google's Trillion Word Corpus, a corpus that is over 1 trillion words in length[48]. To eliminate non-words that were included in this list (such as "f", "gp", and "ooo"), we excluded words composed of 3 or fewer characters if they did not appear in the 'Oxford 5000' list. After supplementing each of these three vocabularies with the words from the original 1152-word vocabulary that were not already included, the three finalized vocabularies contained 3303, 5249, and 9170 words (these sizes are given in the same order that the vocabularies were introduced).

For each vocabulary, we retrained the n-gram language model used during the beam-search procedure with n-grams that were valid under the new vocabulary (Method S4) and used the larger vocabulary during the beam search. We then simulated the sentence-spelling experiments offline using the same hyperparameters that were used during real-time testing.

### Statistics and reproducibility

**Statistical analyses.** The statistical tests used in this work are all described in the figure captions and text. In brief, we used two-sided Wilcoxon Rank-Sum tests to compare any two groups of observations. When the observations were paired, we instead used a two-sided Wilcoxon signed-rank test. We used Holm-Bonferroni correction for comparisons in which the underlying neural data were not independent of each other. We considered P-values <0.01 as significant. We computed P-values for Spearman rank correlations using permutation testing. For each permutation, we randomly shuffled one group of observations and then determined the correlation. We computed the P-value as the fraction of permutations that had a correlation value with a larger magnitude than the Spearman rank correlation computed on the non-shuffled observations. For any confidence intervals around a reported metric, we used a bootstrap approach to estimate the 99% confidence interval. On each iteration (of a total of 2000 iterations), we randomly sampled the data (such as accuracy per cross-validation fold) with replacement and calculated the desired metric (such as the median). The confidence interval was then computed on this distribution of the bootstrapped metric. We used SciPy (version 1.5.4) during statistical testing.

**Reproducibility of experiments.** Because this is a pilot study with a single participant, further work is required to definitively determine if the current approach is reproducible with other participants.

**Data exclusions.** During the copy-typing condition of the sentence-spelling task, the participant was instructed to attempt to silently spell each intended sentence regardless of how accurate the decoded sentence displayed as feedback was. However, during a small number of trials, the participant self-reported making a mistake (for example, by using the wrong code word or forgetting his place in the sentence) and sometimes stopped his attempt. This mostly occurred during initial sentence-spelling sessions while he was still getting accustomed to the interface. To focus on evaluating the performance of our system rather than the participant's performance, we excluded these trials (13 trials out of 163 total trials) from performance-evaluation analyses, and we had the participant attempt to spell the sentences in these trials again in subsequent sessions to maintain the desired amount of trials during performance evaluation (2 trials for each of the 75 unique sentences).

Including these rejected sentences when evaluating performance metrics only modestly increased the median CER and WER observed during real-time spelling to 8.52% (99% CI [3.20, 15.1]) and 13.75% (99% CI [8.71, 29.9]), respectively.

During the conversational condition of the sentence-spelling task, trials were rejected if the participant self-reported making a mistake (as in the copy-typing condition) or if an intended word was outside of the 1152 word vocabulary. For some blocks, the participant indicated that he forgot one of his intended responses when we asked him to report the intended response after the block concluded. Because there was no ground truth for this conversational task condition, we were unable to use the trial for analysis. Of 39 original conversational sentence-spelling trials, the participant got lost on 2 trials, tried to use an out-of-vocabulary word during 6 trials, and forgot the ground-truth sentence during 3 trials (leaving 28 trials for performance evaluation). Incorporating blocks where the participant used intended words outside of the vocabulary only modestly raised CER and WER to median values of 15.7% (99% CI [6.25, 30.4]) and 17.6%, (99% CI [12.5, 45.5]) respectively.

### Reporting summary
Further information on research design is available in the Nature Research Reporting Summary linked to this article.

## Data availability
Relevant data are available under restricted access per the guidelines from our clinical-trial protocol which allow us to share de-identified data with researchers at other institutions but precludes us from making all of the data publicly available. Access can be obtained upon reasonable request. Requests for relevant data can be made to Dr. Edward Chang (edward.chang@ucsf.edu). Responses can be expected within 3 weeks. Any provided data should be kept confidential and should not be shared with others unless approval to do so is obtained from Dr. Chang. The participant has requested to remain anonymous; as a result, information that could identify him is not included in this article and will not be included in any shared data. The source data to re-create the manuscript figures (including accuracies, statistical values, and cross-validation accuracies) are provided with this publication in the associated GitHub repository: github.com/ChangLabUcsf/silent_spelling. Source data are provided with this paper.

## Code availability
The code to train the models, use model predictions from neural data to spell sentences, and recreate all of the figures is available at: github.com/ChangLabUcsf/silent_spelling.

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

## Acknowledgements

We are indebted to our participant Bravo-1 for his tireless dedication to the research project. We also thank members of Karunesh Ganguly's lab for help with the clinical study, Todd Dubnicoff for video editing, Kenneth Probst for illustrations, Nick Halper and Kian Torab for hardware support, members of the Chang Lab for feedback, Viv Her and Clarence Pang for administrative support, and the participant's caregivers for logistic support. For this work, the National Institutes of Health (grant NIH U01 DC018671-01A1) and William K. Bowes, Jr. Foundation supported authors S.L.M., J.R.L., D.A.M., M.E.D., M.P.S., K.T.L., J.C., G.K.A., and E.F.C. Authors A.T.C. and K.G. did not have relevant funding for this work.

## Author contributions

S.L.M. designed and trained the neural classifier, developed real-time classification, language-modeling, and beam-search approaches and software, and developed the offline classification, spelling-simulation, and neural-feature analyses. J.R.L. designed and trained the real-time speech detection model, performed nearest-class distance and evoked-signal analyses, performed statistical assessments, and contributed to the neural-feature analyses. D.A.M. managed and coordinated the research project and designed and implemented the real-time software infrastructure used to collect data and enable real-time sentence spelling. SLM. and J.R.L. generated figures. S.L.M., J.R.L., and D.A.M. designed the spelling process. D.A.M. and M.E.D. designed the graphical user interface for the spelling process. S.L.M., J.R.L., D.A.M., and E.F.C. prepared the manuscript with input from other authors. S.L.M., J.R.L., D.A.M., M.E.D., M.P.S., K.T.L., and J.C. helped collect the data, and, along with G.K.A., were involved in methodological design. M.P.S., A.T.C., K.G., and E.F.C. performed regulatory and clinical supervision. E.F.C. conceived, designed, and supervised the clinical trial.

## Competing interests

S.L.M., J.R.L., D.A.M., and E.F.C. are inventors on a pending provisional patent application that is directly relevant to the neural-decoding approach used in this work. G.K.A and E.F.C are inventors on patent application PCT/US2020/028926, D.A.M. and E.F.C. are inventors on patent application PCT/US2020/043706 and E.F.C. is an inventor on patent US9905239B2 which are broadly relevant to the neural-decoding approach in this work. The remaining authors declare no competing interests.
