## [Peer Review File · Nature Communications]

Generalizable spelling using a speech neuroprosthesis in an individual with severe limb and vocal paralysisREVIEWER COMMENTS

Reviewer #1 (Remarks to the Author):

The manuscript aimed to demonstrate real-time, neural prosthesis-based generalizable spelling in a paralyzed person. One participant with anarthria (severe dysarthria with intact cognition) was recruited and a 128-channel ECoG was implanted in the sensorimotor cortex on the left hemisphere. A neural network was trained by imagining isolated NATO code words (representing letters) and then used to decode silently spelled sentences, where spaces were automatically inserted. Some highlighted results include median character error rate was as low as 6.13% with a rate of 29.4 characters per minute, which is faster than using typing-based AAC device by the same participant. The authors also studied a few paradigms with interesting results obtained (e.g., low- vs high-frequency and overt vs covert speech). I think the results are a significant advancement towards the final clinical goal of speech BCI. The manuscript is well written. Details of the experiment, modeling, and results are provided and clear. The idea of using NATO code words is interesting and was proven to be higher performance (than isolated letters). A few moderate to minor improvements are suggested before it can be published (see details below).

Below are just two general questions before other specific comments.

First, was there a test or evaluation on the participant's articulation (e.g., tongue and lip motion). If the participant can still move tongue, jaw, and lips in a consistent way in producing silent speech, a silent speech interface (non-invasive) would help and there won't be a need for a neurosurgeon to use BCI. It can be briefly discussed in the Discussion section.

J. A. Gonzalez-Lopez, A. Gomez-Alanis, J. M. Martín Doñas, J. L. Pérez-Córdoba and A. M. Gomez, "Silent Speech Interfaces for Speech Restoration: A Review," in *IEEE Access*, vol. 8, pp. 177995-178021, 2020, doi: 10.1109/ACCESS.2020.3026579.

Second, did or could the participant use any feedback (correct or incorrect spelling on the screen) during sentence spelling?

Introduction

Line 46-47, references are needed here about the literature on neural speech decoding or synthesis. Different neural modalities have been attempted non-invasive modalities (e.g., EEG and MEG), although their performances are not as good as invasive ones. Other neural speech decoding work using invasive modalities by the same authors and other groups can be cited here as well. For example,

Cooney, C., Folli, R., & Coyle, D. H. (2021). A bimodal deep learning architecture for EEG-fNIRS decoding of overt and imagined speech. *IEEE Transactions on Bio-medical Engineering* (in press). Advance online publication. <https://doi.org/10.1109/TBME.2021.3132861>

Dash, D., Ferrari, P., & Wang, J. (2020). Decoding imagined and spoken phrases from non-invasive neural (MEG) signals, *Frontiers in Neuroscience*, 14(290), 1-15.

Herff, C. Krusienski, D. J., Kubben, P. (2020). The Potential of Stereotactic-EEG for Brain-Computer Interfaces: Current Progress and Future Directions, *Front. Neuosci.* <https://doi.org/10.3389/fnins.2020.00123>

Wilson, G. H., Stavisky, S. D., Willett, F. R., Avansino, D. T., Kelemen, J. N., Hochberg, L. R.,

Henderson, J. M., Druckmann, S., & Shenoy, K. V. (2020). Decoding spoken English from intracortical electrode arrays in dorsal precentral gyrus. *Journal of neural engineering*, 17(6), 066007. <https://doi.org/10.1088/1741-2552/abbfef>

In the paragraph starting with line 54, a few more sentences would be helpful to make it clear what are the improvement (besides larger vocabulary) this study brought over ref [16] (e.g., [16] is not real-time).

Results

Line 305, how to apply PCA on the temporal content to measure temporal dimensionality?

Discussion

In the paragraph starting with line 492, a brief comparison with the performance with the study on imagining hand-writing motor (Ref [6]), which would be more insightful. For example, [6] may be faster, but as mentioned earlier, users prefer “speech” communication, which is more natural.

In the paragraph starting with line 523, other work using MEG also verified low frequency information is useful in decoding speech for patients with mild dysarthria (due to ALS).

Dash, D., Ferrari, P., Hernandez, A., Heitzman, D., Austin, S., & Wang, J. (2020). Neural speech decoding for amyotrophic lateral sclerosis, *Proc. Interspeech*, pp. 2782 - 2786.

It's wonderful the ECoG has been implanted for 128 weeks. How long can it stay in the brain? Does the participant need to go back to the clinic regularly to check if the implant is in good shape?

That would be great to have a discussion on potential challenges to apply this technology to individuals with locked-in syndrome in the end of the Discussion section. For example, what improvements are needed to be tested with individuals with locked-in syndrome and it can be used in daily life.

Methods

Line 636, it's good to state it explicitly the area of the implant (sensorimotor cortex area, according to ref [16]), rather than just vaguely saying “the areas association with speech processing”.

Below are a few additional minor comments.

Line 274, is Fig. S5 actually Fig. 3A?

Line 527-556 and other places, citations should be in numbers.

Supplemental Materials

Line 476, the refs [18] and [24] seem inappropriate here. Please check.

Reviewer #2 (Remarks to the Author):

This is a compelling, technically impressive further demonstration that brain-to-speech approaches are feasible. In that, this paper extends and complements earlier reports by this group on (to my understanding) this patient, who thankfully continues to contribute in extraordinary ways to the field's progress in this regard.

What I am less convinced about is the uniquely new contribution this manuscript/these data will provide.

-- In my reading, the notable extension over and above previous analyses here lies in the patient performing word sub-articulation of NATO phonetic alphabet words (rather than spelling single letters like other brain-based spellers might have been approaching this, and like this group has done before as well). The superiority of the NATO spelling approach is, to my reading, clearly demonstrated here. The paper then goes on to demonstrate the (title-giving) generalisation as per modelling the performance on a much larger and realistic vocabulary ("offline simulation").

-- The utility of combining low- and high-frequency activity for better classification performance has face validity and is reassuring to be demonstrated here, but to my knowledge has been demonstrated before (eg by the Howard/Nourski lab). The decoding of an intended hand-motor signal is innovative and clever, While I am also not certain about the novelty of this. Happy to stand corrected on both of these grounds.

In sum, the manuscript holds some impressive heavy-lifting and data wrangling, but there can be no doubt that the contribution of the paper is in neuroengineering, and less so in neurobiological or neurocognitive insight.

This is either a feature or a bug of this study, depending on editorial viewpoint.

The authors themselves also state the relative incrementality of the here collected findings. They conclude «Together, these findings provide further insight into the neural representations underlying silent-speech generation with a paralyzed person and expand the demonstrated viability of speech neuroprosthetics as a practical approach for communication restoration.»

As outlined above, a revised version of this paper should, in my view, focus first on a transparent exposition of where and how this study overlaps with vs goes beyond the previous reports on this patient.

Secondly, the offline generalisation to new vocabulary struck me as more reassuring than truly a novel finding in its own right. If the authors see this very differently (as the title implies), they might need to make more clear in the manuscript why and how this is the case.

TECHNICAL COMMENTS:

Related to my concerns about novelty and incrementality, the paper should be much more transparent in outlining and highlighting the overlap in terms of setup, data, participant with earlier reports by the lab.

I would strongly suggest these data to be made publicly available, in the spirit of open science also extending to the ECoG/neuroengineering community.

In Figure 1, why is gamma referred to as "activity" and low-frequency as "signal" – both are power time courses, I assume? Related, these two time courses for illustration only really have the same frequency content themselves? That is, should the low-F track not also look more lowpass than the high-F one? (More a question of not to mislead readers who are familiar with those kind of signals)

The statistical approach is solid (as is the modelling using cross-validation) and the results are for most very clean-cut; I thus see no need to quibble over details here. Maybe only this: It's not really state of the art any longer to report only p-values (however small they might be in most instances here) as measures of effect size. I would strongly recommend to provide Mann Whitney z values or, even better, point and interval estimates for all measures/differences of relevance.

Reviewer 1 comments

9. ***Was there a test or evaluation on the participant's articulation (e.g., tongue and lip motion). If the participant can still move tongue, jaw, and lips in a consistent way in producing silent speech, a silent speech interface (non-invasive) would help and there won't be a need for a neurosurgeon to use BCI. It can be briefly discussed in the Discussion section.***

[J. A. Gonzalez-Lopez, A. Gomez-Alanis, J. M. Martín Doñas, J. L. Pérez-Córdoba and A. M. Gomez, "Silent Speech Interfaces for Speech Restoration: A Review," in IEEE Access, vol. 8, pp. 177995-178021, 2020, doi: 10.1109/ACCESS.2020.3026579.]

Thank you for this astute suggestion.

A speech-language pathologist did perform an assessment with the participant. We have included a summary of the results of this assessment in a new Supplementary Note in the revised submission. The contents of that new Note are provided below.

Regarding the potential use of a non-brain-based silent-speech interface (SSI), we think that it's possible that that could work with this participant to some degree, but of course this would need to be tested before its efficacy could be truly known. We use the term "non-brain-based SSI" here to mean non-invasive SSIs that don't involve brain recordings, disambiguating them from other potential non-invasive SSIs, such as EEG-based approaches. Given the participant's articulatory deficits, we hypothesize that non-brain-based SSIs wouldn't be very fruitful for the current participant. As for brain-based non-invasive SSIs, using recording methodologies such as EEG or MEG, we have not seen speech-decoding studies based on these technologies achieve the same levels of performance as invasive ECoG-based studies (see [Makin et al. 2020, Sun et al. 2020, Anumanchipalli et al. 2019]).

We do want to emphasize here that the long-term goal of our clinical trial (of which this work is a part) is to develop technology that could be usable by patients regardless of their residual vocal-tract control. The performance of non-brain-based SSIs is presumably positively correlated with level of residual vocal-tract control, meaning that patients with total or near-total loss of vocal-tract control would not be able to use those

systems. Although we acknowledge that assessing the performance of the same participant with non-brain-based SSIs could be insightful and could enrich the present findings, unfortunately our clinical-trial protocol doesn't currently enable us to test an EMG setup with the participant.

Given our desire to focus the manuscript and the Discussion on brain-based methods, we have decided not to add content about non-brain-based SSIs to the manuscript. If you feel strongly about including this content, let us know and we can reconsider.

Note S1. Assessment of the participant's articulatory inventory

The participant was diagnosed with anarthria by a certified speech-language pathologist. This section contains a summary of the results from an articulatory-inventory assessment with the participant conducted by this speech-language pathologist. The participant's articulation was characterized through a battery of tests.

An oral-mechanism test was used to assess gross movement of the jaw, tongue, and lips. The outcomes of this test indicated significant deficits in the articulatory performance of the participant. Residual jaw movement was relatively reliable, but all movements were very effortful and slow. The participant was also unable to produce lip rounding or puckering, and he could not maintain lip closure. Tongue performance was very poor and he could only produce a limited range of slow extension and elevation tongue movements.

A perceptual dysarthria assessment was used to measure the reliability of syllable-sequence production during overt-speech attempts. The participant was unable to produce multisyllabic sequences, with increasing breakdowns as the number of syllables per word increased and as the number of words per utterance increased. The participant's speech movements were characterized predominantly by spastic features, resulting in articulatory imprecision and slow rates of speech (about 1-2 syllables per breath).

A speech-articulation task was used to characterize performance on each phonetic consonant. The participant could only reliably produce nasals (/m/, /n/, /ng/) and a single palatal liquid (/y/). Other consonants included errors in voicing or were replaced with more reliable consonants; nasals would replace stops, and liquids would replace fricatives and affricates. Additionally, the participant's spasticity restricts the range and speed of articulatory motions, contributing to poor control of airflow and phonation.

- 10. *Did or could the participant use any feedback (correct or incorrect spelling on the screen) during sentence spelling?***

We did not have the participant correct sentences during spelling, since the language model could correct sentences and change the position of spaces later in the sentence as more character probabilities limited the possible sentences. This is similar to other brain-computer interfaces (Willett et. al. 2021).

From lines 120-122 of the original submission: We instructed the participant to continue spelling even if there were mistakes in the displayed sentence, since the beam search could correct the mistakes after receiving more predictions.

To clarify that the participant was unable to modify the sentence both during and after sentence spelling, we added the following sentence to the Methods section:

We did not implement functionality to allow the participant to retroactively alter the predicted sentence, although the language model could alter previously predicted words in a sentence after receiving additional character predictions.

11. ***Line 46-47, references are needed here about the literature on neural speech decoding or synthesis. Different neural modalities have been attempted non-invasive modalities (e.g., EEG and MEG), although their performances are not as good as invasive ones. Other neural speech decoding work using invasive modalities by the same authors and other groups can be cited here as well. For example,***
[Cooney, C., Folli, R., & Coyle, D. H. (2021). A bimodal deep learning architecture for EEG-fNIRS decoding of overt and imagined speech. IEEE Transactions on Bio-medical Engineering (in press). Advance online publication. <https://doi.org/10.1109/TBME.2021.3132861>]
[Dash, D., Ferrari, P., & Wang, J. (2020). Decoding imagined and spoken phrases from non-invasive neural (MEG) signals, Frontiers in Neuroscience, 14(290), 1-15.]
[Herff, C. Krusienski, D. J., Kubben, P. (2020). The Potential of Stereotactic-EEG for Brain-Computer Interfaces: Current Progress and Future Directions, Front. Neuosci. <https://doi.org/10.3389/fnins.2020.00123>]
[Wilson, G. H., Stavisky, S. D., Willett, F. R., Avansino, D. T., Kelemen, J. N., Hochberg, L. R., Henderson, J. M., Druckmann, S., & Shenoy, K. V. (2020). Decoding spoken English from intracortical electrode arrays in dorsal precentral gyrus. Journal of neural engineering, 17(6), 066007. <https://doi.org/10.1088/1741-2552/abbfef>]

We agree that including additional references on non-invasive modalities and invasive modalities would help add additional background for readers. We have included the suggested literature as additional references in line 51 of the original manuscript. For a reference on stereo-EEG, we exchanged the suggested paper (***[Herff, C. Krusienski, D. J., Kubben, P. (2020). The Potential of Stereotactic-EEG for Brain-Computer Interfaces: Current Progress and Future Directions, Front. Neuosci.***

<https://doi.org/10.3389/fnins.2020.00123>) with [Angrick, M., Ottenhoff, M., Goulis, S., Colon, A. J., Wagner, L., Krusienski, D. J., ... & Herff, C. (2021). *Speech Synthesis from Stereotactic EEG using an Electrode Shaft Dependent Multi-Input Convolutional Neural Network Approach*. In *2021 43rd Annual International Conference of the IEEE Engineering in Medicine & Biology Society (EMBC)* (pp. 6045-6048). <https://doi.org/10.1109/EMBC46164.2021.9629711>], since the first paper talks solely about the potential for Stereotactic EEG to be used for speech decoding, but the updated reference is able to demonstrate it.

12. ***In the paragraph starting with line 54, a few more sentences would be helpful to make it clear what are the improvements (besides larger vocabulary) this study brought over ref [16] (e.g., [16] is not real-time).***

Thank you for raising this suggestion. We agree with you, and the other reviewer, that the manuscript would benefit from a clearer description of the advantages of the current work compared to the previous one. We think that two major advantages that can be highlighted in the Introduction section is the generalizability to larger vocabulary and also the use of silent speech attempts instead of overt ones. The point about generalizability is already present in the manuscript, but we have added a sentence in this paragraph to emphasize the point about silently attempted speech compared to overtly attempted speech.

Additionally, please view our response to comment 25 (from the other reviewer) for how we further modified the Discussion section to describe the improvements of this work compared to the prior one.

Separately, in this prior work the participant was controlling the neuroprosthesis by attempting to speak aloud, making it unclear if the approach would be viable for potential users who cannot produce any vocal output whatsoever.

13. ***Line 305, how to apply PCA on the temporal content to measure temporal dimensionality?***

We agree that additional detail on how this was done is necessary and have added additional details in the “Performance evaluation” subsection of the Methods section of the main text.

We then arranged this into a matrix with dimensionality $N \times TC$, where N is the number of features (128 for HGA and for LFS; 256 for HGA+LFS), T is the number of time points in each 2.5-second window, and C is the number of NATO code words (26), by concatenating the trial-averaged activity for each feature. We then performed principal component analysis along the feature dimension of this matrix. Additionally, we arranged

the trial-averaged data for each code word into a matrix with dimensionality T x NC. We then performed principal component analysis along the temporal dimension.

14. ***In the paragraph starting with line 492, a brief comparison with the performance with the study on imagining hand-writing motor (Ref [6]), which would be more insightful. For example, [6] may be faster, but as mentioned earlier, users prefer “speech” communication, which is more natural.***

Thank you for this great suggestion; we agree that this is a good point to explicitly state in the discussion and have modified that paragraph to include it.

BCI performance using penetrating microelectrode arrays in motor cortex has steadily improved over the past 20 years [Serruya et. al. 2002, Gilja et. al. 2012, Kawala-Sterniuk et. al. 2021], recently achieving spelling rates as high as 90 characters per minute with a single participant [Willett et. al. 2021], although this participant was able to speak normally. Our results extend the list of immediately practical and clinically viable control modalities for spelling-BCI applications to include silently attempted speech with an implanted ECoG array, which may be preferred for daily use by some patients due to the relative naturalness of speech [Branco et. al. 2021] and may be more chronically robust across patients through the use of less invasive, non-penetrating electrode arrays with broader cortical coverage.

15. ***In the paragraph starting with line 523, other work using MEG also verified low frequency information is useful in decoding speech for patients with mild dysarthria (due to ALS). [Dash, D., Ferrari, P., Hernandez, A., Heitzman, D., Austin, S., & Wang, J. (2020). Neural speech decoding for amyotrophic lateral sclerosis, Proc. Interspeech, pp. 2782 - 2786.]***

Thank you for pointing this relevant study out. We have included its findings, along with the findings of other studies that used low-frequency activity for neural decoding in the introduction to the results section titled: “Discriminatory content in high-gamma activity and low-frequency signals”.

Previous efforts to decode speech from brain activity have typically relied on content in the high-gamma frequency range (between 70–170 Hz, but exact boundaries vary) during decoding [Herff et. al. 2014, Makin et. al. 2012, Moses et. al. 2019]. However, recent studies have demonstrated that low-frequency content (between 0–40 Hz) can also be used for spoken- and imagined-speech decoding [Mugler et. al. 2014, Sun et. al. 2015, Dash et. al. 2020, Proix et. al. 2022, Anumanchipalli et. al. 2019] although the differences in the discriminatory information contained in each frequency range remain poorly understood.

16. ***It's wonderful the ECoG has been implanted for 128 weeks. How long can it stay in the brain? Does the participant need to go back to the clinic regularly to check if the implant is in good shape?***

This is a great question. ECoG arrays (rather than a strip of 4-10 electrodes) implanted in humans are still predominantly used in the acute research setting and we have an investigational device exemption for this trial to determine exactly what you're asking—the safety and efficacy of chronically implanted ECoG for BCI. ECoG strips, however, are chronically implanted much more widely for the treatment of epilepsy, dystonia, and other disorders. These devices have been shown to be safely implanted for upwards of 9 years (<https://pubmed.ncbi.nlm.nih.gov/32690786/>). Assuming no infections or other adverse medical events, our clinical trial covers an implant duration of 5 years.

Our participant doesn't go to the clinic regularly to check on the device, rather we have several protocols in place to identify, off-site, whether a clinic visit is necessary. This includes a cleaning protocol that we perform before and after each recording visit to prevent infection and maintain a healthy implant site, a health survey completed at the start of each recording session to assess any possible issues (such as headaches or other incidents that may occur while we are not recording). We also monitor the impedances of each electrode on the array at the start of each recording session and these measures have remained relatively stable. The participant additionally meets with a clinician on our team each month for a detailed check-up on his health and the health of the implant. Our clinical trial includes protocols, such as a CT scan, if a problem is detected with the implant.

17. ***That would be great to have a discussion on potential challenges to apply this technology to individuals with locked-in syndrome in the end of the Discussion section. For example, what improvements are needed to be tested with individuals with locked-in syndrome and it can be used in daily life.***

Thanks for this suggestion. We think there are two things here that can be addressed separately.

First is how the technology can be validated with individuals with locked-in syndrome. We hypothesize that the approach should be successful to some extent if the patient's speech-motor cortex is intact. This will of course require more testing, both through potential enrollment of individuals with locked-in syndrome in our clinical trial and through other research groups also investigating this. We do think that the current methods described in this work would be applicable to testing with these individuals. We've updated some text in the Discussion section to make this clearer.

Second is about how the approach can be used in daily life. Because of the percutaneous connector which passes through the skull and scalp, clinical and regulatory considerations dictate that a team member be present when connecting and disconnecting the system. Therefore, autonomous usage isn't possible in the current setup. Additionally, we did not develop software to integrate the spelling approach with a word processor or other applications. We added a sentence in the final paragraph of the Discussion section to mention what steps can be taken to move towards autonomous usage of a system based on this spelling approach.

These findings illustrate the viability of attempted-speech control for individuals with complete vocal-tract paralysis (such as those with locked-in syndrome), although future studies with these individuals are required to further our understanding of the neural differences between overt-speech attempts, silent-speech attempts, and purely imagined speech as well as how specific medical conditions might affect these differences. We expect that the approaches described here, including recording methodology, task design, and modeling techniques, would be appropriate for both speech-related neuroscientific investigations and BCI development with patients regardless of the severity of their vocal-tract paralysis, assuming that their speech-motor cortices are still intact and that they are mentally capable of attempting to speak.

Although clinical and regulatory guidelines concerning the implanted percutaneous connector prevented the participant from being able to use the current spelling system independently, development of a fully implantable ECoG array and a software application to integrate the decoding pipeline with an operating system's accessibility features could allow for autonomous usage.

18. ***Line 636, it's good to state it explicitly the area of the implant (sensorimotor cortex area, according to ref [16]), rather than just vaguely saying "the areas association with speech processing".***

Thank you for bringing this to our attention. We've modified this sentence to specifically list the areas.

The array was surgically implanted on the pial surface of the left hemisphere of the brain over cortical regions associated with speech production, including the dorsal posterior aspect of the inferior frontal gyrus, the posterior aspect of the middle frontal gyrus, the precentral gyrus, and the anterior aspect of the postcentral gyrus [Bouchard et. al. 2013, Chartier et. al. 2018, Guenther et. al. 2016].

19. ***Line 274, is Fig. S5 actually Fig. 3A?***

We apologize for the confusion here. Fig. S5 is the confusion matrix, whose overall

classification accuracy is also reported in Fig. 3A. We've amended the reference to Fig. S5 to also include Fig. 3A, which we hope makes this more clear.

Models using only LFS demonstrated higher code-word classification accuracy than models using only HGA, and models using HGA+LFS outperformed the other two models ($P < 0.001$ for all comparisons, two-sided Wilcoxon Rank-Sum test with 3-way Holm-Bonferroni correction; Fig. 3A, S4, Table S3), achieving a median classification accuracy of 54.2% (99% CI [51.6, 56.2], Fig. 3A, S5).

20. *Line 527-556 and other places, citations should be in numbers.*

Thank you for pointing this out. Citations have been updated to be consistent throughout the text.

21. *In the supplement, Line 476, the refs [18] and [24] seem inappropriate here. Please check.*

Thank you for raising this concern. Reference [24] contains the training details for the GPT-2 language model. The Distil GPT-2 model is entirely based on this language model, and it is trained via the process of 'knowledge distillation' to recreate the predictions and behavior of this model, so we believe including this citation is important. However, the case for reference [18] is a little more complicated. In short, the creator of DistilGPT-2 did not submit a publication introducing this software. Instead, it is based on the techniques used during the creation of DistilBERT (which is described in a publication; this is reference [18]) and primarily described in a blog post. To prevent other readers from being reasonably confused at this, we decided to also cite this blog post. We hope that the result is clearer. The bibliography in the supplement is updated to include this blog post (provided below), and we've updated the citations in that sentence to include this. Please note that the reference numbers in the revised supplement are not identical to the reference numbers in the original supplement.

[Victor Sanh. Smaller, faster, cheaper, lighter: Introducing DistilBERT, a distilled version of BERT. 2019. url: <https://medium.com/huggingface/distilbert-8cf3380435b5>]

Reviewer 2 comments

22. *In my reading, the notable extension over and above previous analyses here lies in the patient performing word sub-articulation of NATO phonetic alphabet words (rather than spelling single letters like other brain-based spellers might have been approaching this, and like this group has done before as well). The superiority of the NATO spelling approach is, to my reading, clearly demonstrated here. The*

paper then goes on to demonstrate the (title-giving) generalisation as per modelling the performance on a much larger and realistic vocabulary (“offline simulation”).

Thank you for this feedback. We first want to point out that, to our knowledge, this is the first demonstration of speech-controlled spelling in a paralyzed person. We have not done this before, and neither has any other group (again, to the best of our knowledge). Please let us know if you do not believe this to be the case.

Additionally, this will be the first demonstration we’ve done with a paralyzed person involving sub-vocal (silent) natural-speech attempts. This is one aspect of the present work that differentiates it from what we’ve done before, as in that previous work the participant was attempting to speak aloud. As we describe in the Discussion section, we think that this is an important distinction and a noteworthy advance, because it shows that residual vocalizations are not necessary for speech-based control of a BCI (and further supports the viability of this approach for eventual clinical applications). As before, to the best of our knowledge we aren’t aware of any other publication which has shown that silently attempted word productions can be decoded from a person with anarthria.

Thank you for your comment regarding our NATO approach and the spelling generalization. We discuss generalizability a bit more in our response to comment 25; please refer to that comment for more details.

23. The utility of combining low- and high-frequency activity for better classification performance has face validity and is reassuring to be demonstrated here, but to my knowledge has been demonstrated before (eg by the Howard/Nourski lab). The decoding of an intended hand-motor signal is innovative and clever, While I am also not certain about the novelty of this. Happy to stand corrected on both of these grounds.

While the combination of low- and high-frequency activity for improved decoding of speech production has been demonstrated before [Anumanchipalli et al. 2019 ; Sun et. al. 2020], it was not clear how the LFS signals improved decoding over just using HGA. Our paper is the first to provide a clearer understanding of this phenomenon. In addition to quantifying the benefits of using HGA and LFS over just HGA, we showed that LFS provides higher temporal and spatial resolution than HGA (Fig 3F, 3G), and has more diffuse electrode contributions than HGA alone (Fig 3B, 3D). We also show that in our case, LFS was actually more discriminable than HGA (Fig 3A). This result is surprising given that many recent papers use solely HGA for neural speech decoding (Makin et. al. 2020, Herff et. al. 2015), and ours is the first paper to demonstrate this effect. We also show that combining HGA and LFS leads to improved decoding performance (Fig 3A) and increased spatial variability (Fig 3F). These contributions demonstrate the added benefits of using LFS in addition to HGA, which were previously unknown. We have

emphasized this is in the first paragraph of the “Discriminatory content in high-gamma activity and low-frequency signals” section, which is referenced in the response to comment 15. Furthermore, to emphasize that combining low- and high- frequency activity improves speech decoding, we revised our accuracies for results examining the effects of HGA vs LFS (results shown in Fig 3) and comparing overt vs silent speech (Fig 5) to only evaluate the decoding accuracy on NATO codeword attempts (previously, these accuracies took into account accuracy from decoding the attempted hand squeeze). Results remained statistically significant, and the accuracy using the NATO codeword attempts and the attempted hand squeeze is still reported in Supplementary Figure S5 (alongside a full confusion matrix).

The decoding of a hand-motor signal in conjunction with speech production attempts has not been demonstrated before. Previous studies have shown that speech production and hand-motor activity can be decoded in the dorsal motor cortex (Wilson et. al. 2020; Willett et. al. 2021), but there has been no demonstration of using both modalities at once to control a BCI. Here, we showed that the hand-motor signal could be discriminated from neural activity during attempted speech with high fidelity (98.43% accuracy). This shows that using both hand-motor signals and speech to drive a BCI can produce highly discriminable signals, which are critical for high performance.

We have updated the discussion to emphasize the novelty of combining the two modalities.

In addition to enabling spatial coverage over the lateral speech-motor cortical brain regions, the implanted ECoG array also provided simultaneous access to neural populations in the hand-motor (“hand knob”) cortical area that is typically implicated during executed or attempted hand movements [Gerardin et. al. 2000]. Our approach is the first to combine the two cortical areas to control a BCI. This ultimately enabled our participant to use an attempted hand movement, which was reliably detectable and highly discriminable from silent-speech attempts with 98.43% classification accuracy (99% CI [95.31, 99.22]), to indicate when he was finished spelling any particular sentence.

24. In sum, the manuscript holds some impressive heavy-lifting and data wrangling, but there can be no doubt that the contribution of the paper is in neuroengineering, and less so in neurobiological or neurocognitive insight. This is either a feature or a bug of this study, depending on editorial viewpoint. The authors themselves also state the relative incrementality of the here collected findings. They conclude «Together, these findings provide further insight into the neural representations underlying silent-speech generation with a paralyzed person and expand the demonstrated viability of speech neuroprosthetics as a practical approach for communication restoration.»

Thank you for including this opinion about the framing of our work in your review.

First, we definitely agree with you that this is more of a neuroengineering contribution than a neuroscientific one. Although there are of course some coarse-level neuroscientific takeaways, including the role of low-frequency signals in speech decoding and some basic characterizations of the neural representations of silent speech with an anarthric person, it was our intent to frame the work primarily as neuroengineering. In addressing comment 17, we revised the text in the Discussion in a way that we think further emphasizes the need for true neuroscientific investigations in future studies. Please refer to that comment for more details.

Second, we do want to respectfully push back a little bit on your statement concerning the “relative incrementality” of the present findings. As stated above, we do agree that our work is more about neuroengineering than neuroscience, but we disagree with the notion that this inherently makes the work “incremental.” We think that many of the key results from this work are novel advances, including the ability to decode silent-speech attempts, leverage multiple frequency ranges and behavioral modalities (speech and hand-motor), and achieve good decoding performance after over 2 years since the device was implanted, all with a paralyzed and anarthric person. Furthermore, we believe that the methods and results described herein will be of great interest to other BCI researchers as well as stakeholders in the technology (including patients with paralysis that could benefit from communication BCIs), given that they expand what is shown to be possible with a speech-based communication BCI. Please see our response to comment 25, which we think further describes the novel advances of the current work.

- 25. *As outlined above, a revised version of this paper should, in my view, focus first on a transparent exposition of where and how this study overlaps with vs goes beyond the previous reports on this patient. Secondly, the offline generalisation to new vocabulary struck me as more reassuring than truly a novel finding in its own right. If the authors see this very differently (as the title implies), they might need to make more clear in the manuscript why and how this is the case.***

Thank you for this feedback. Regarding your first point, we agree that it is important to convey how these results extend beyond our previous work and that we can revise the manuscript to convey this more clearly. We think that the Introduction section does sufficiently describe our previous work and its limitations. But, we think the Discussion section should more clearly summarize the key advances in this work compared to the previous one. We have revised the first paragraph in the Discussion section to make this clearer. Please also view our response to comment 12 (from the other reviewer) in how we've modified the Introduction section to also emphasize the advances to readers.

Regarding your second point, we do want to push back slightly on the notion that only the offline results demonstrated generalizability. We think that it is reasonable to consider a vocabulary of over 1000 words (real-time) as generalizable in the context of BCIs for communication. For patients who could benefit from this technology, we think that the variety of thoughts that can be communicated with this vocabulary size can be considered generalizable, at least to a certain degree. But, we of course agree that going up to 9000 words (offline) is even more generalizable, especially considering that it is more than sufficient for basic fluency. We've revised some text in the Discussion section to make this more explicit.

These results significantly expand our previous word-decoding findings with the same participant [Moses, Metzger, Liu et. al. 2021] by enabling completely silent control, leveraging both high- and low-frequency ECoG features, including a non-speech motor command to finalize sentences, facilitating large-vocabulary sentence decoding through spelling, and demonstrating continued stability of the relevant cortical activity beyond 128 weeks since device implantation.

In post-hoc analyses, we showed that decoding performance improved as more linguistic information was incorporated into the spelling pipeline. This information helped facilitate real-time decoding with a 1,152-word vocabulary, allowing for a wide variety of general and clinically relevant sentences as possible outputs. Furthermore, through offline simulations, we validated this spelling approach with vocabularies containing over 9,000 common English words, which exceeds the estimated lexical-size threshold for basic fluency and enables general communication.

26. ***Related to my concerns about novelty and incrementality, the paper should be much more transparent in outlining and highlighting the overlap in terms of setup, data, participant with earlier reports by the lab.***

Thank you for this suggestion. Given our modifications as part of our response to comment 25, we hope that some of your concerns are alleviated. Additionally, throughout the Methods section we do emphasize that the hardware and some aspects of the software are consistent with the approaches we used in our previous work [Moses, Metzger, Liu et al. 2021] (see, for example, the "Neural implant" and "Data acquisition and preprocessing" subsections). The participant is indeed the same from that work (as stated in the "Participant" subsection), but we do agree with your point that we can be more explicit about any potential data overlap with the previous study. To be clear, none of the data from any previous work with the participant were used here, and we've modified the text in the "Participant" subsection to make this clear.

This participant has participated in previous studies as part of this clinical trial [Moses, Metzger, Liu et. al. 2021, Silversmith et. al. 2020], although neural data from those

studies were not used in the present study.

27. *I would strongly suggest these data to be made publicly available, in the spirit of open science also extending to the ECoG/neuroengineering community.*

We definitely agree that the data would be of interest to the neuroengineering community and also want to support open science where we can. Unfortunately, our clinical protocol dictates that we keep our data secure and prevents open-sourcing the neural data. We are considering trying to get approval from the regulatory bodies that oversee our work to alleviate this restriction and allow us to open-source our data, but we can't make any guarantees about that at this time.

However, our protocol does contain the following allowance: "De-identified electrophysiological data may be shared with other researchers at other institutions." This is what we plan on doing if this work is accepted for publication. Specifically, we want to make the data available to other researchers upon reasonable request, so that other experts in the field can gain access to the de-identified data. This is reflected in the "Data and Code availability" section.

28. *In Figure 1, why is gamma referred to as "activity" and low-frequency as "signal" – both are power time courses, I assume? Related, these two time courses for illustration only really have the same frequency content themselves? That is, should the low-F track not also look more lowpass than the high-F one? (More a question of not to mislead readers who are familiar with those kind of signals)*

We apologize for the confusion here and thank you for bringing it to our attention. To clarify, the time course we refer to as "high-gamma activity" is the analytic amplitude of the high-gamma frequency range of the raw neural signal. That is, we take the raw signal at 1 kHz and compute the Hilbert transform for 8 log-spaced frequency bands within the high-gamma range (70 to 150 Hz). We take the analytic amplitude for each of these bands and average them together, yielding our final high-gamma activity feature at a 200 Hz sampling rate (which we then z-score and may also further downsample, depending on the model).

The feature we refer to as "low-frequency signal" is the low-passed raw neural signal at 200 Hz. We chose to use the term "signal" here because this is more close to an observed signal than the high-gamma feature is. The high-gamma feature is processed to only include the analytic amplitude, so we felt it would be inaccurate to call this feature a "signal" as it is more an operation on a signal.

To the question of why both appear to be equally "lowpass", the high-gamma analytic amplitude moves more slowly than the full high-gamma signal (that is, if the signal still contains amplitude and phase components). By design, the analytic amplitude operation

smooths a signal and in fact, the frequency content of the high-gamma feature is mostly in the lower frequencies of the analytic amplitude. Therefore, in Figure 1, both features can appear equally smooth.

We hope that explanation clarifies why we chose those terms, but please let us know if these terms are still not clear enough.

29. *The statistical approach is solid (as is the modeling using cross-validation) and the results are for most very clean-cut; I thus see no need to quibble over details here. Maybe only this: It's not really state of the art any longer to report only p-values (however small they might be in most instances here) as measures of effect size. I would strongly recommend to provide Mann Whitney z values or, even better, point and interval estimates for all measures/differences of relevance.*

Thank you for this comment, we have included these values as part of our revised paper. Since we are providing the z-values instead of the U-statistics for each group, we have accordingly changed the name of the statistical tests to "Wilcoxon Rank-Sum" from "Mann-Whitney U".

REVIEWER COMMENTS

Reviewer #1 (Remarks to the Author):

The revised manuscript has addressed all my concerns and clarified all my questions. I appreciate the authors for their great effort in this revision. I think the manuscript is ready to be published in its current form.

Reviewer #2 (Remarks to the Author):

I am content with the very insightful replies the authors gave and the changes they have performed accordingly. From my reviewer point of view, the current version deserves publication in Nat Comms as is.

Jonas Obleser

Response to reviewer comments

9. ***The revised manuscript has addressed all my concerns and clarified all my questions. I appreciate the authors for their great effort in this revision. I think the manuscript is ready to be published in its current form.***

We thank the reviewer for their consideration of our revisions and for their approval of the revised manuscript.

10. ***I am content with the very insightful replies the authors gave and the changes they have performed accordingly. From my reviewer point of view, the current version deserves publication in Nat Comms as is.***

We thank the reviewer for their consideration and approval of our responses and revisions and for their recommendation for publication.